# Warm Cells, Hot Mitochondria: Achievements and Problems of Ultralocal Thermometry

**DOI:** 10.3390/ijms242316955

**Published:** 2023-11-29

**Authors:** Alexey G. Kruglov, Alexey M. Romshin, Anna B. Nikiforova, Arina Plotnikova, Igor I. Vlasov

**Affiliations:** 1Institute of Theoretical and Experimental Biophysics of the Russian Academy of Sciences, 142290 Pushchino, Russia; nikiforanna@yandex.ru; 2Prokhorov General Physics Institute of the Russian Academy of Sciences, 119991 Moscow, Russia; vlasov@nsc.gpi.ru; 3Institute for Physics and Engineering in Biomedicine, National Research Nuclear University MEPhI (Moscow Engineering Physics Institute MEPhI), 115409 Moscow, Russia; arina.plotnikova.eug@gmail.com

**Keywords:** nanothermometry, sensors, bioenergetics, cell thermogenesis, mitochondria, heat production

## Abstract

Temperature is a crucial regulator of the rate and direction of biochemical reactions and cell processes. The recent data indicating the presence of local thermal gradients associated with the sites of high-rate thermogenesis, on the one hand, demonstrate the possibility for the existence of “thermal signaling” in a cell and, on the other, are criticized on the basis of thermodynamic calculations and models. Here, we review the main thermometric techniques and sensors developed for the determination of temperature inside living cells and diverse intracellular compartments. A comparative analysis is conducted of the results obtained using these methods for the cytosol, nucleus, endo-/sarcoplasmic reticulum, and mitochondria, as well as their biological consistency. Special attention is given to the limitations, possible sources of errors and ambiguities of the sensor’s responses. The issue of biological temperature limits in cells and organelles is considered. It is concluded that the elaboration of experimental protocols for ultralocal temperature measurements that take into account both the characteristics of biological systems, as well as the properties and limitations of each type of sensor is of critical importance for the generation of reliable results and further progress in this field.

## 1. Introduction

Temperature is rightfully one of the most significant regulators of cellular activity, which governs the chemical reaction rates in various compartments [1,2,3,4] and determines the viscosity of biological membranes and their phase transitions [5,6,7,8], the rate of diffusion [9], the transcription of genes [10], and the frequency and rate of conformational changes in macromolecules [11], as well as their stability and plasticity [12].

Since the occurrence of biochemical reactions is possible with a decrease in the Gibbs free energy, and since an increase in thermal entropy in biological systems is the least harmful for maintaining homeostasis, most reactions produce heat or consume energy (in the form of ATP), some of which is thermally dissipated. In fact, local heat dissipation may cause the creation of nanoscale thermal gradients between cellular compartments, resulting in a global temperature increase in the bodies of fairly large organisms with a high metabolic rate. Therefore, an experimental disclosure of local temperature gradients within distinct compartments of a single living cell is one of the most challenging issues for ultralocal thermometry. The released heat in the local region can initiate or stimulate a cascade of site-specific reactions, giving rise to so-called “thermal signaling”, when the temperature variation in one part of a living cell acts as a transmitted signal for another [13].

For the last two decades, in order to clarify the existence of intracellular local thermal gradients and validate the consistency of the concept of “thermal signaling”, plenty of methods have been developed and adapted for temperature detection in diverse cellular compartments at the micro- and nanoscale. Among them are techniques using thermocouples [14,15,16], microcantilevers [17], microdiodes [18], nanodiamonds [19,20,21,22,23], quantum dots [24,25,26,27], fluorescent dyes and proteins [28,29,30,31,32,33,34,35], polymeric [36,37,38,39,40,41,42,43,44] and metallic nanoparticles [45,46], rare-earth ions [47,48,49], and others [50,51,52,53,54]. Resolving the problem of local thermal gradients and substantiating the concept of “thermal signaling” would allow one to start the study of the physiological and pathological role of these phenomena, as well as to develop approaches to the modulation and control of cellular processes.

However, the results of measurements of the local intracellular and cellular temperature obtained with the use of different methods and thermal sensors are fairly contradictory. This may be due to artifacts, inappropriate signal calibration protocols, and the fact that at least some methods and sensors suffer from interfering interactions with the intracellular content, low stability, dependence of the signal on nonthermal intracellular factors, and poor, uneven, or poorly controlled delivery of the sensor to the cell or organelle.

This review highlights various methods and sensors for local and ultralocal thermometry in cells and cellular compartments, their advantages, limitations, and possible sources of errors and ambiguities. The review analyzes the data on the temperature of various types of cells and cell organelles in different metabolic states, obtained using these methods. The biological consistency of the data is critically evaluated. In addition, data on biological temperature limits within cells and organelles, which, when exceeded, leads to irreversible damage to intracellular structures and cell death, are considered.

## 2. Review Chapters

An ideal thermosensor should possess a high spatiotemporal resolution to be capable of catching the temperature variations at a specific site of a living cell or organelle over a long period of time. Thermal sensing should be achieved with a minimum perturbation of biological objects and meet the spatial requirements for targeting specific cellular compartments. The existing thermometric approaches applicable for the quantification of thermogenic activity in biological systems can be subdivided by their principle of operation into two classes—nonoptical and optical. While the former require direct physical contact with cells/organelles, the latter read out the temperature fully optically allowing convenient noninvasive recordings. Below, we review the techniques of both classes that have already demonstrated their feasibility to study the temperature distribution in biological objects from single organelles to cells. For more information about all state-of-the-art nanothermometry techniques and their working principle, see, e.g., [55,56,57].

### 2.1. Nonoptical Methods

*Thermocouples.* A thermocouple (TC) is a traditional approach to control the temperature with remarkable accuracy [14,15,16,54]. The operation principle is based on the thermal dependence of the electron energy distribution in metals. At the electrical junction of dissimilar metals, temperature variations lead to the formation of an electromotive force across the temperature gradient and, consequently, to the appearance of a voltage difference between “cold” ends of metals. The ultimate thermoelectric power is quantitatively characterized with the Seebeck coefficient, which is the quotient of voltage per temperature changes. Since the TC only measures the voltage difference between dissimilar metals, it does not reflect the absolute temperature and therefore requires either preliminary calibration in a thermostat or the use of a reference bath with a known temperature (e.g., ambient conditions).

An adaptation of TCs in biological avenues requires the miniaturization of thermoelectric junctions up to the nanoscale. Recently, reports were published about a W-PU-Pt (wolfram–polyurethane–platinum) probe of conical shape with a submicron size as small as 100 nm [15,16] and its successful application for intracellular temperature measurements by direct mechanical penetration through the cell membrane. The Seebeck coefficient of 6.5 μV/${}^{{}^{\circ}}$C and the 0.05 ${}^{{}^{\circ}}$C thermal resolution allowed the detection of 0.5–1 ${}^{{}^{\circ}}$C temperature changes in MLE-12 and U251 cell lines [15,16].

Despite the significant accuracy and sensitivity of thermocouples, there are some limitations of their use for the characterization of biological heat sources. First, a region of detection is not limited by the electrical junction of metals. Due to high thermal conductivity, the external metal layer can detect heat production from thermal sources that are far from the alloy point, thereby extending the detection volume. Furthermore, a TC is not protected from variations in other external parameters such as local electromagnetic fields and pH, which can potentially lead to an ambiguous interpretation of voltage changes. Finally, the complex fabrication process of TC hampers its ubiquitous utilization.

*Microcantilevers.* A bimaterial microcantilever is another nonoptical instrument that exhibits a potential for the detection of biological thermogenic activity. It uses the effect of micromechanical bending of a probe to quantify temperature changes. The materials of the microcantilever are chosen to have different coefficients of thermal expansion [17]. Under “hot” conditions, each material expands differently, resulting in the bending of the probe. The amount of heat production is then evaluated from the degree of the displacement of the tip of the microcantilever, which proportionally depends on temperature with a slope of 9.15 μm/${}^{{}^{\circ}}$C. The minimum achievable temperature resolution of this sensor is 0.7 mK. This technique is suitable for working on an extracellular microscale level with relatively big biological objects of size comparable to the probe, e.g., tissue or cells. Thus, it was demonstrated that several brown adipose cells produced a heat of 0.4 ${}^{{}^{\circ}}$C under norepinephrine application [17].

*Microdiode.* Yamada et al. [18] suggested an electric microsensor based on the pn junction of a diode for single-cell-level measurements. It consisted of two materials, n-type and p-type silicon, which were brought into close contact. The resistance of the pn diode decreased with the increasing in situ temperature, which made it possible to read absolute temperature variations at a certain bias voltage applied. To operate with a single cell, the authors designed a sensor as a microbridge between two isolated microfluidic chambers allowing to catch a brown adipose cell (BA) from one side and detect thermogenic activity on the other side at the pn junction point. The thermal sensitivity of 1.1 mK made it possible to observe a 0.1 ${}^{{}^{\circ}}$C heating of the BA after the application of 1 µM norepinephrine.

Although the area of active thermal control of the above-mentioned methods is localized at the micro-/nanoscale, they inherently have a bulky geometry that could diminish the absolute thermal power of the nanoscale thermal source due to the additional spreading and scattering of heat in the bulk substance of the probe. In physical terms, these methods constitute nonclosed thermodynamic systems with parasitic heat capacity and thermal conductivity, which, therefore, complicates their utilization for reliably visualizing the temperature of specific compartments within the cell.

### 2.2. Optical Methods

Optical methods form a vast class that attracts attention due to the ability to monitor temperature without direct perturbation of the biological object. The majority of them use the temperature dependence of luminescence—the ability to emit photons from a specific substance, occurring primarily from the excited electronic state populated by an external source of light, e.g., laser radiation. The electronic structure involved in photon emission changes in the presence of temperature gradients results in an alteration of luminescence parameters such as intensity, spectral position, linewidth, lineshape, and lifetime.

*Fluorescent dyes and genetically encoded proteins.* A fluorescent dye represents a small organic compound, usually of molecular origin, that emits light. The fluorescence takes place when the molecule is excited into the upper electronic state (excited state) with a certain vibrational mode and relaxes back to an electronic state with a lower energy, simultaneously emitting a photon. Typically, a specific fluorescent dye has an individual absorption (the part of the spectrum where the molecule effectively absorbs the light) and emission bands that are usually located in the visible region of the spectrum. In most cases, an increase in temperature leads to a de-excitation of the molecule through either phonon-mediated electron transfer to a higher vibrational state from which the radiation relaxation is not favorable or through multiphonon nonradiative relaxation to the ground electronic state. Therefore, the variety of fluorescent parameters are used for temperature registration—from intensity and lineshape to lifetime and ratiometricity between emission intensities.

The small size of the organic molecule allows it to read out the temperature noninvasively inside the cell by its accumulation in particular cellular compartments, e.g., mitochondria [32,33,58] and endo-/sarcoplasmic reticulum (ER/SR) [30,59]. Moreover, it is possible to design site-specific molecules targeting predefined cell compartments [33].

Another subclass of molecular probes is genetically encoded fluorescent proteins (GFP, gTEMP) [28,29,30]. The operation principle is quite similar to that of the above-mentioned dyes; as a result, thermal changes can be monitored by intensity and spectral position. In addition, it was demonstrated that conformational changes between neutral and anionic states of tsGFP could be used as a ratiometric thermosensor with an accuracy of ∼0.2 ${}^{{}^{\circ}}$C, as could be seen from [29]. The nonsynthetic nature of GFP allows them to measure the temperature noninvasively in intact cells [29]. Moreover, GFP can be directed to a specific organelle in a cell through the attachment of chemical sequences that target a specific domain, and its expression can be controlled by promoters [29,30].

It is worth noting that most fluorescent probes of the current group have a spectral dependence on third-party parameters, among which are pH, electromagnetic fields, viscosity, concentration, refractive index, etc. Also, the photobleaching of organic dyes is a stumbling stone, precluding their continuous use in long-term protocols of temperature detection.

*Nanogels and polymeric nanoparticles.* The potential use of fluorescent nanogel as an intracellular nanothermometer was first introduced by Gota et al. [43]. The concept is based on the combination of the thermoresponsive polymeric unit poly(*N*-isopropylacrylamide) (pNIPAM) and the water-sensitive fluorophore *N*-{2-[(7-*N*,*N*-dimethylaminosulfonyl)-2,1,3 benzoxadiazol-4-yl](methyl)amino}ethyl-*N*-methylacrylamide (DBD-AA). At low temperatures, the polymeric unit assumes an extended structure with the hydration of the amide linkages, resulting in a swelling nanogel. The DBD-AA fluorescence is then quenched by neighboring water molecules at the interior of the nanogel. At higher temperatures, the pNIPAM shrinks with a simultaneous release of H_2_O molecules from the nanogel volume and a subsequent strong rise in the DBD-AA fluorescence. The DBD-AA intensity increases drastically only in the range of reduced temperatures (from 27 ${}^{{}^{\circ}}$C to 33 ${}^{{}^{\circ}}$C [43]) where the conformational changes occur, leading to a maximum achievable temperature resolution of 0.3 ${}^{{}^{\circ}}$C [43]. Despite this, the authors succeeded in monitoring the temperature inside COS7-cells by a microinjection of nanogel, although the large size (∼50 nm) and the low hydrophilicity of nanogel prevented its homogeneous distribution along the cell compartments. In later works, the use of smaller polymeric nanoparticles (9 nm) of different composition with a better thermal resolution of 0.1 ${}^{{}^{\circ}}$C was demonstrated [39], which allowed the mapping of intracellular temperature variations between cytosol, nucleus, centrosome, and mitochondria. Furthermore, the cationic subunit serving as an intermediate link between polymer and fluorophore was reported to facilitate the internalization of the cell membrane [40,44], paving the way for convenient and safe cell-permeable thermometry. Also, the polymeric unit could bind two fluorophores with different environmental sensitivity, giving rise to the ratiometric thermometry with improved resolution (up to 0.01 ${}^{{}^{\circ}}$C) and extended the thermal operating range (25–45 ${}^{{}^{\circ}}$C) [40].

*Quantum dots.* Quantum dots (QDs) are colloidal semiconductor crystals whose carriers are clamped in three spatial dimensions. The fluorescence process in a QD is closely related to the appearance of the resonant motion of electrons by virtue of spatial constraint. Therefore, the absorption and emission bands depend on the size, shape, and nature of nanoparticles and can be tuned to a desirable region (from ultraviolet to infrared). QDs, in general, are much brighter than organic dyes and possess a high quantum yield and photostability, which allow them to be used as robust intracellular sensors. The main mechanism of temperature dependence is attributed to the scattering of the phonons on charge carriers and energy transfer processes from bulk to surface nonradiative states that affect the intensity, bandwidth, and spectral position of fluorescence lines [26,60]. There are several experimental works that use a shift of spectral position [26] and the ratio of individual parts of the emission spectrum [24,25,27] as the quantity of thermogenic activity. Both approaches reveal a detection accuracy of ∼0.5 ${}^{{}^{\circ}}$C [24,26] and demonstrate the ability of intracellular temperature detection at the nanoscale. A possible weak point of QDs is their sparse localization within the cell and their cytotoxicity, which limits in vivo applications.

*Nanodiamonds.* Diamond is an optically transparent and chemically inert dielectric material. The wide band gap allows the formation of fluorescent color centers (so called “artificial atoms”) with an isolated energy structure. Their optical properties are strongly determined by the type of impurity embedded inside the diamond during the synthesis. The best-known impurities, nitrogen and silicon, can form the nitrogen-vacancy (NV) and silicon-vacancy (SiV) color centers, respectively. The former emits photons in the visible range and is found in two different charge states: neutral (NV^0^) with the zero-phonon line (ZPL) at 575 nm and negative with ZPL at 638 nm (NV^−^). Both NV^0^ and NV^−^ possess a wide phonon sideband owing to the axial trigonal symmetry in the diamond lattice. Instead, the SiV center has a narrow (∼6 nm) ZPL at 738 nm in the first biological optical transparency window and a near-uniform absorption starting from 450 nm. The thermal sensitivity of these color centers is provided primarily by the electron–phonon scattering processes which take place in the diamond crystal lattice. The size of diamond particles can be precisely controlled from micrometers to several nanometers, providing versatile opportunities in the high-resolution thermal imaging of biological objects in both extra-/intracellular ways.

As with the aforementioned approaches, it is possible to measure the temperature in biological systems using the conventional spectral parameters of fluorescence. For example, recently, it was reported about a temperature elevation up to 20 ${}^{{}^{\circ}}$C close to isolated mouse brain mitochondria using the spectral position of the SiV emission line [23]. However, there is another powerful way of thermal readout with an NV center, which is called optically detected magnetic resonance (ODMR). The energy structure of the NV center consists of ground and excited states, each of which undergoes a temperature-dependent Zeeman splitting in the absence of magnetic field (so called zero-field splitting). Using resonant microwave radiation, it is possible to initialize the spin state with a simultaneous decrease in the fluorescence. Thus, by performing a microwave frequency sweep, one can obtain ODMR spectra with minima corresponding to the resonant frequency and shifting with the temperature. Of note, recent experiments have revealed the ability of ODMR for intracellular thermal sensing in human embryonic fibroblasts under physiological conditions [20] and in primary culture of neurons under a stimulated spiking activity [23].

In general, nanodiamonds constitute highly photostable and robust thermometers. They are not sensitive to the chemical composition of the local environment and are devoid of such disadvantages as photo-/thermobleaching and cell toxicity. As the size of particles can be reduced up to the molecular level [61], NDs appear to be one of the most promising candidates for nanothermometric applications in biology.

*Rare-earth ions.* Rare-earth ions (REIs, or lanthanide ions) represent the core of another thermometric group. Owing to an abundant energy structure, REIs have optical electrons on the distant orbitals (D, F, etc.) that do not participate in the formation of chemical bonds. However, being free ions, REIs per se are not optically active due to selection rules on the orbital and energy numbers. Toward that end, they are usually embedded either into a crystalline [47] and polymeric [48] environment or different chelates (e.g., TTA) [49]. Under these conditions, the energy levels of the ions slightly shift, which results in a partial mixing of the states with different parities. As a consequence, electric dipole transitions between/inside distant orbitals accompanied by emissions in visible and near infra-red ranges become permitted. The thermal dependence of REIs’ emission lies in the increased role of nonradiative energy transfer and thermal redistribution between the closest excited states [47]. In case of chelated REIs, radiationless energy transfer from excited ions to polydentate ligands provides the sensitivity of the probe to the temperature by means of the fluorescence intensity [49].

There are two ways to measure the temperature using REIs. The former is based on the analysis of the Stokes fluorescence, giving the opportunity to detect thermal variations by spectral parameters with a resolution in temperature of ∼0.5 ${}^{{}^{\circ}}$C, as could be estimated from [49]. The latter utilizes the energy transfer between two different ions to reach an up-conversion effect—an emission at shorter wavelengths than the excitation. The efficiency of this process is also coupled with temperature, ensuring a temperature resolution of ∼1 ${}^{{}^{\circ}}$C in biological conditions [47].

Noteworthy, the fluorescence of chelated REIs such as Eu-TTA is reported to be dependent on nonthermal parameters (pH, viscosity, etc.) [49]. Photobleaching is also one of the headaches of this approach [49]. The REIs in solid state are more photostable without sensitivity to environment parameters for bulky ions, but except the ions that are located on the surface of the solid which are capable of perceiving external local nonthermal fluctuations.

*Metallic nanoparticles*. Metallic nanoclusters (MNC) less than 2 nm in size may fluoresce in visible range [45,46]. While the exact mechanism of fluorescence is not yet clearly understood, it was found that the lifetime and intensity were reversely responsive to the environmental temperature. Using FLIM of gold nanoparticles endocytosed in HeLa cells, Shang et al. mapped the intracellular temperature throughout distinct compartments in cytoplasm [46]. The operation range of this probe was between 15 and 45 ${}^{{}^{\circ}}$C, whereas the temperature resolution was 0.1–0.3 ${}^{{}^{\circ}}$C [46]. Wang et al. [45] reported that a fluorescence intensity of a thermometer based on cuprum nanoparticles also had a capacity to evaluate thermogenic activity inside the MC3T3-E1 cell. In their approach, the upper limit of the operation range was shifted to 70 ${}^{{}^{\circ}}$C, while the thermal resolution remained the same [45]. Generally, metal nanoparticles possess excellent photo- and colloidal stability and exhibit good biocompatibility owing to their submolecular size. However, there are some limitations related to (1) the simultaneous heating of MNC during the exposure of the excitation source and (2) a possible sensitivity to external environmental parameters (e.g., cuprum MNCs are susceptible to oxidation).

Eventually, all aforementioned methods have limitations in biological thermal imaging that are related to either the environmental sensitivity of nonthermal parameters or the indirect initiation of biological activity (pharmacological and therapeutic actions). Below, the aspects of heat-mediated biological processes are considered.

### 2.3. Local Intracellular Temperature

In living cells, myriads of chemical reactions constantly produce warmth. This would lead to the rapid heating of cells to critical temperatures if the heat were not removed and dissipated in the environment. Reactions that release warmth occur in all compartments of the cell. However, in some compartments, thermogenesis is particularly intensive, which is believed to lead to the formation of local thermal gradients under certain physiological or pathological conditions [62,63,64,65]. According to current views, these compartments primarily include mitochondria [23,62,63], endo-/sarcoplasmic reticulum (ER/SR) [29,34,35,59,66,67], and the nucleus [28,37,39,40]. The high rate of heat generation in mitochondria is associated with the functioning of the respiratory chain, especially under conditions of uncoupling of respiration and oxidative phosphorylation [68,69]. It is believed that sarcoplasmic and endoplasmic reticulum Ca^2+^ ATPase (SERCA) is responsible for the generation of heat and formation of temperature gradients in reticular structures [70,71,72]. A high heat production in the nucleus and the presence of a temperature gradient between the nucleus and the cytoplasm may be connected either with the intensive processes of the DNA replication and transcription [73,74] or with a high SERCA activity in the external leaflet of the nuclear envelope [70,71,75]. Cytosol, lysosomes, endosomes, lipid droplets, etc. are much less often considered as thermal gradient-generating compartments, presumably because of the relatively low rate of generation of warmth. In contrast, the plasma membrane, which produces a lot of heat (for example, up to 30% of cellular ATP is hydrolyzed by the plasma membrane Na^+^/K^+^ ATPase [76]), is apparently too extended to establish a considerable thermal gradient. According to theoretical considerations and mathematical models, whole cells should be incapable of forming significant temperature gradients in relation to the surrounding medium [64,65]. However, the existence of gradients, though very variable in their magnitude, was shown experimentally with different cell types, using different methods and for different metabolic conditions. Below, the question regarding the existence of temperature gradients in cells and various intracellular structures and the possible biological and methodological reasons for discrepancies in experimental data on these gradients are considered in more detail.

#### 2.3.1. Whole Cell, Cell Surface, and Cytosol

Table 1 summarizes the data on the measurements of the augmentation in the cell/cell surface temperature (T_cell_) or cytosolic temperature (T_cyto_) in relation to the temperature of the surrounding medium (T_med_), obtained by different experimental approaches. The table includes data from studies that quantified “resting” T_cell_(T_cyto_) or ΔT (T_cell_(T_cyto_) − T_med_) after a certain stimulus. As follows from the data presented, only some instruments allow a measure of the temperature gradient between the cell and the environment (microcantilever [17], fluorescent nanodiamonds [22], nanoparticle-based thermometer [77], quantum dots [24], fluorescent nanosheets [66], and high-molecular weight polymers (e.g., R-CFPT) [78]). Therefore, the data on the T_cell_(T_cyto_) of “resting”, unstimulated cells are few, and at the same time, they are quite contradictory both for undifferentiated (cancer and embryonic cell lines) and for terminally differentiated BAs.

In particular, the T_cell_(T_cyto_) of HeLa and HEK293 cells was measured to be the same as [77], close to (+0.12–0.19 ${}^{{}^{\circ}}$C) [66], or much higher (+1.5 ± 0.5 ${}^{{}^{\circ}}$C) [22] than T_med_ when detected by a nanoparticle-based ratiometric thermometer, fluorescent nanosheets, and fluorescent nanodiamonds, respectively. Using commercial quantum dots, the T_cyto_ of SH-SY5Y cells was found to be considerably elevated in comparison with T_med_: +0.8 ${}^{{}^{\circ}}$C in neurites and +2.4 ${}^{{}^{\circ}}$C in the cell body [24]. Similarly, in BAs and pre-BAs, the assessment using R-CFPT showed a large temperature gradient between the cell and the environment +2.3 ± 0.2 ${}^{{}^{\circ}}$C (pre-BA) and +4.4 ± 0.2 ${}^{{}^{\circ}}$C (BA) [78]. In contrast, measurement with a microcantilever did not reveal the existence of any gradient [17].

These contradictions appear to be related both to the morphology of terminally differentiated and undifferentiated cells and to the peculiarities of the measurement methods. It is quite obvious that the microanatomy of an undifferentiated cancer cell or fibroblast-like cell differs considerably from that of BAs or cardiomyocyte. In the first case, the cytosol is filled with organelles and membrane structures quite diffusely, which, on the one hand, facilitates the exchange of metabolites with the environment and between organelles, and on the other hand, eases the dissipation of heat by the cell (Figure 1a). Nonetheless, even in undifferentiated cells, the specific resistance to heat transfer is approximately six times higher than that of water [79]. In the terminally differentiated BAs, huge lipid droplets fill the entire space of the cytosol and displace other organelles into the narrow gaps between the droplets, which should further reduce the overall cellular thermal conductivity and hamper the dissipation of heat from the cytosol, causing its local overheating (Figure 1b). Accordingly, the cytosolic sensor R-CFPT indicated almost a 4.5 ${}^{{}^{\circ}}$C rise in T_cyto_ in BAs [78], while the microcantilever located at a distance of 2–7 μm from the cell surface and shielded by several layers of lipids did not record the changes in T_cell_ [17]. In differentiated myocytes, a branched membrane (ER, T-tubules) and ordered fibrillar structures (sarcomers) can play the role of intracellular thermal insulators (Figure 1c). The remarkable difference in the cytosolic temperature of HeLa cells, measured with nanosized sensors, namely Eu-TTA/Rhod 101-containing nanoparticles and nanodiamonds, may be connected either with their different cytosolic localization or with a relatively low accuracy of both methods (∼1 ${}^{{}^{\circ}}$C) [22,77].

**Table 1 ijms-24-16955-t001:** The temperature of the cell, cell surface, and cytosol and its stimulus-dependent changes.

Cell Type *	Assay **	Thermosensor ***	Stimulus ****	$T_{\mathrm{med}}/\Delta T_{\mathrm{cyto}}$, $T_{0}/\Delta T_{\mathrm{st}}$ *****	Ref.
COS-7	Fluorescence microscopy	1, nanogel polyNIPAM-MBAM-DBD-AA	100 μM FCCP	28 ${}^{{}^{\circ}}$C/+0.45 (0–2) ${}^{{}^{\circ}}$C	[43]
COS-7	TCSPC system-based FLIM	FPT, polyNNPAM- SPA- DBD-AA	(NA μM) FCCP (30 min)	30 ${}^{{}^{\circ}}$C/+1.02 ± 0.17 ${}^{{}^{\circ}}$C average; +1–4 ${}^{{}^{\circ}}$C around mitochondria	[39]
NIH/3T3	Fluorescence microscopy	QD, quantum dots (QD655, Invitrogen, Waltham, MA, USA)	1 μM ionomycin-Ca^2+^	37 ${}^{{}^{\circ}}$C/+1.84 ± 0.27 ${}^{{}^{\circ}}$C (from −2 to +8 ${}^{{}^{\circ}}$C)	[26]
HeLa, COS-7, NIH/3T3	FLIM	AP4-FPT, nanogel, polyNNPAM- APTMA- DBThD-AA	10 mM CCCP	30 ${}^{{}^{\circ}}$C/+1.57 ± 1.41 ${}^{{}^{\circ}}$C (HeLa)	[37]
HeLa	Confocal fluorescence microscopy, single-photon detection	RFP, two-component ratiometric fluorescent polymer PolyNIPAM-co-NBDAA:PolyNIPAM-co-RhBAM (100:1)	(NA μM) FCCP	33.3 ${}^{{}^{\circ}}$C/+2.0–2.4 ${}^{{}^{\circ}}$C	[42]
HeLa	ODMR technique	FND, fluorescent nanodiamonds	No stimulus	32 ${}^{{}^{\circ}}$C/+1.5 ± 0.5 ${}^{{}^{\circ}}$C	[22]
HeLa	Confocal microscopy	tsGFP1, GFP-TlpA fusion protein	10 μM CCCP	37 ${}^{{}^{\circ}}$C/$\Delta T_{st}\to 0$	[29]
HeLa	Confocal microscopy	RNT, ratiometric nanothermometer, Eu-TTA and rhodamine 101 embedded in PMMA nanoparticle covered by PAH	No stimulusIonomycin (high)	$T_{cyto}=T_{med}$ 37 ${}^{{}^{\circ}}$C/(+2 (0.5–3.5) ${}^{{}^{\circ}}$C (transient peak within ∼100–150 s)	[77]
HeLa	Fluorescence microscopy	Pipette filled with Eu-TTA	2 μM ionomycin 2 μM thapsigargin 1 h prior to ionomycin	22 ${}^{{}^{\circ}}$C/≤+1 ${}^{{}^{\circ}}$C Suppression of warming	[49]
HEK293T	TCSPC system-based FLIM	FPT, linear cationic fluorescent polymeric thermometer, polyNNPAM- APTMA- DBD-AA	Aβ42 (for 24 h) Aβ42 + MJ040X FCCP Aβ42 + MJ040X + FCCP	37 ${}^{{}^{\circ}}$C/+2.8 ± 0.6 ${}^{{}^{\circ}}$C / $\Delta T_{st}\to 0$ /+10.0 ± 1.2 ${}^{{}^{\circ}}$C /+10.0 ± 1.2 ${}^{{}^{\circ}}$C	[80]
J774A.1 HEK293T	Thermography in cell suspension	Thermography catheter	+100 μM NaCN (10 min) UCP2 overexpression (>95% cells)	23 ${}^{{}^{\circ}}$C/+0.13 ± 0.048 ${}^{{}^{\circ}}$C (J774A.1) 23 ${}^{{}^{\circ}}$C/+0.16 ± 0.068 ${}^{{}^{\circ}}$C (HEK293T)	[81]
HeLa HEK293 BA Rat neonatal cardiomyocytes Rat hippocampal neurons	Phase contrast (DIC) fluorescent microscopy	Nanosheets fluorescent thermometer containing EuTTA and rhodamine 101 (thickness ∼ 50 nm)	No stimulus 2 μM ionomycin Type 1 ryanodine receptor mutation 10 μM CCCP 2 Hz electrical stimulation 0.25 Hz electrical stimulation	36 ± 1 ${}^{{}^{\circ}}$C/+ 0.12–0.19 ${}^{{}^{\circ}}$C (HEK293) /$\Delta T_{st}\to 0$ (±0.15 ${}^{{}^{\circ}}$C , HeLa) /≤+0.1 ${}^{{}^{\circ}}$C (HEK293) /$\Delta T_{st}\to 0$ (±0.1 ${}^{{}^{\circ}}$C , cardiomyocytes, BA) /$\Delta T_{st}\to 0$ (±0.01 ${}^{{}^{\circ}}$C , cardiomyocytes) /$\Delta T_{st}\to 0$ (±0.03 ${}^{{}^{\circ}}$C , neurons)	[66]
C2C12 myoblasts and differentiated myotubes	Time-domain FLIM, frequency-domain FLIM, confocal laser scanning microscopy	mCherry, fluorescent protein	1 mM caffeine	37 ${}^{{}^{\circ}}$C/−0.07 ± 0.18 ${}^{{}^{\circ}}$C	[67]
BA, pre-BA	Confocal microscopy	R-CFPT, ratiometric–cationic fluorescent polymeric thermometer, polyNNPAM-APTMA-DBThD-AA-BODIPY-AA	No stimulus 10 μM FCCP 30 μM FCCP NE CL316.243	30 ${}^{{}^{\circ}}$C/+2.3 ± 0.2 ${}^{{}^{\circ}}$C (pre-BA) /+4.4 ± 0.2 ${}^{{}^{\circ}}$C (BA)/+1.5 ${}^{{}^{\circ}}$C (10 min, BA); (20–30 min, pre-BA) /+2.5 and +3.5 ${}^{{}^{\circ}}$C (10 and 30 min, BA) /+1.25 ± 0.25 ${}^{{}^{\circ}}$C /+1.39 ± 0.38 ${}^{{}^{\circ}}$C (at 31 min, BA)	[78]
BA, BA-ASK1-KO	Fluorescence microscopy	1, nanogel polyNIPAM- MBAM- DBD-AA	0.5 μM CL316.243 ASK1-KO + CL316.243	NA ${}^{{}^{\circ}}$C/ +1.29 ${}^{{}^{\circ}}$C (at 30 min) NA ${}^{{}^{\circ}}$C/ +0.52 ${}^{{}^{\circ}}$C (at 30 min)	[82]
BA	Bright-field microscopy	Microcantilever (cell–sensor distance ∼2–7 μm)	No stimulus 1 μM NE (30 min)	25 ± 1 ${}^{{}^{\circ}}$C/ $\Delta T_{cyto}\to 0$ /+0.217 ± 0.120 ${}^{{}^{\circ}}$C	[17]
BA	Microvoltmetry	Si pn junction diode thermal sensor	1 μM NE	23 ${}^{{}^{\circ}}$C/+0.1 ${}^{{}^{\circ}}$C (in 5–20 min interval)	[18]
BA	TCSPC system-based FLIM	DTG, lipid droplets thermo green, BODIPY-n-undecanoyl fusion	1 mM ISO (50 min)	37 ${}^{{}^{\circ}}$C/−0.24 ± 1.0 ${}^{{}^{\circ}}$C	[35]
*Aplysia californica* neurons	Microvoltmetry	Au/Pd thermocouple with silicon nitride cantilever	10 μM BAM15	23 ${}^{{}^{\circ}}$C/+7.5 ± 2.0 ${}^{{}^{\circ}}$C (relaxation within 33 s)/+0.1–0.2 ${}^{{}^{\circ}}$C (external thermometer)	[83]
U251	Microvoltmetry	Thermocouple, W/polyurethane/Pt sandwich	12 μM camptothecin 50 μM doxorubicin	23 ${}^{{}^{\circ}}$C/+ 0.6 ± 0.2 ${}^{{}^{\circ}}$C (30 min) 23 ${}^{{}^{\circ}}$C/+ 0.1 ± 0.1 ${}^{{}^{\circ}}$C (30 min)	[15]
SH-SY5Y	Confocal fluorescent laser scanning microscopy	QD, quantum dots (Qtracker nanocrystals, Invitrogen)	No stimulus 10 μM CCCP	37 ${}^{{}^{\circ}}$C/+0.8 ${}^{{}^{\circ}}$C (neurites), +1.4 ${}^{{}^{\circ}}$C (cell body) /+0.94 ${}^{{}^{\circ}}$C average (−2.5–+5.5 ${}^{{}^{\circ}}$C)	[24]
CHO-K1	TCSPC system-based FLIM	NPs_Eu2, Eu-TTA complex embedded in latex nanoparticles	10–30 μM FCCP	25 ${}^{{}^{\circ}}$C/+3–4 ${}^{{}^{\circ}}$C (endosomes/lysosomes)	[48]

Notes: *—COS-7, African green monkey fibroblast-like kidney cell line expressing SV40 T-antigen; NA, not available; NIH/3T3, mouse fibroblast-like cell line; HeLa, human cervical cancer cells; HEK293T, human embryonic kidney 293 cells containing the SV40 T-antigen; J774A.1, murine macrophages; BA, brown adipocytes; BA-ASK1-KO, brown adipocytes knockout for ASK1 protein kinase; pre-BA, brown adipocyte precursor cells; U251, human malignant glioblastoma; SH-SY5Y, neuroblastoma cell line subcloned from SK-N-SH; CHO-K1, Chinese hamster ovary cells. **—TCSPC, time-correlated single-photon counting; FLIM, fluorescence lifetime imaging microscopy; ODMR, optically detected magnetic resonance. ***—APTMA, 3-(acrylamidopropyl)trimethylammonium; BAM15, 5-*N*,6-*N*-bis(2-fluorophenyl)-[1,2,5]oxadiazolo[3,4-b]pyrazine5,6-diamine; BODIPY-AA, 8-(4-acrylamidophenyl)-4,4-difluoro-1,3,5,7-tetramethyl-4-bora-3a,4a-diaza-s-indacene; DBD-AA, N-{2-[(7-*N*,*N*-dimethylaminosulfonyl)-2,1,3-benzoxadiazol-4-yl](methyl)amino}ethyl-*N*-methylacrylamide; DBThD-AA, N-(2-{[7-(*N*,*N*-dimethylaminosulfonyl)-2,1,3-benzothiadiazol-4-yl]-(methyl)amino}ethyl)-*N*-methylacrylamide; EuTTA, europium (III) thenoyltrifluoroaceton-ate trihydrate; FND, fluorescent nanodiamonds with negatively charged nitrogen-vacancy centers; MBAM, *N*,*N’*-methylenebisacrylamide; NBDAA, 4-(2-acryloylaminoethylamino)-7-nitro-2,1,3-benzoxadiazole; NIPAM, N-Isopropylacrylamide; NNPAM, *N*-n-propylacrylamide; PAH, poly-(allylamine) hydrochloride; PMMA, poly(meth-yl methacrylate); RhB-AM, rhodamine B derivative; SPA, potassium 3-sulfopropyl acrylate; TPP+, triphenylphosphonium. ****—NE, norepinephrine; ISO, isoproterenol. *****—$T_{med}$, $T_{cyto}$, $T_{0}$, and $T_{st}$, the temperature of the medium, cytosol, cytosol before stimulus, and cytosol after stimulus, respectively; $\Delta T_{cyto}=T_{cyto}-T_{med}$; $\Delta T_{st}=T_{st}-T_{0}$.

The set of methods that allow the measure of an increase in T_cell_(T_cyto_) after exposure to any thermogenic stimulus is much richer (Table 1). According to the mechanism of activation of thermogenesis, stimuli can be divided into several groups. First of all, these are uncouplers of respiration and oxidative phosphorylation, which act directly on the inner membrane of mitochondria (FCCP, CCCP, and BAM15) [24,29,37,39,42,43,48,66,78,80,83]. The second group is presented by the β-adrenergic receptor (β-AR) agonists (norepinephrine, isoproterenol, and CL316.243) [17,18,35,78,82], which activate UCP1 in the mitochondria of BAs, causing mitochondrial uncoupling and the acceleration of heat generation. The same effect produces the overexpression of UCP2 in other cell types [81]. The third group is composed of activators of the ATPase activity of SERCA and/or the actin–myosin complex. It includes the Ca^2+^ ionophore ionomycin [26,49,66,77], agonist of ryanodine receptors caffeine [29], and electrical pulses for excitable cells (cardiomyocytes, neuronal cells) [66]. The expression of mutant ryanodine receptors can also elevate T_cyto_ via the increased Ca^2+^ leak from the ER/SR [66,72]. (Below we look into why the attribution of Ca^2+^-dependent heat generation exclusively to SERCA or/and actin-myosin complex is not entirely correct.) The last group includes various damaging agents with diverse targets in cells (Aβ42, NaCN, and inhibitors of DNA topoisomerase I and II) [15,80,81]. However, their thermogenic effects cannot be unambiguously associated with a specific process in the cell.

As follows from the data presented in Table 1, the addition of protonophore uncouplers, FCCP/CCCP, in most cases led to an increase in intracellular temperature (T_cyto_). Measurements with polymeric gels, namely, 1 (here and further are the authors’ designations), FPT, AP4-FPT, RFP, and R-CFPT revealed an increase in T_cyto_ of 0.45, 1, 1.6, 2.0–2.5, 1.5, and 3.5 ${}^{{}^{\circ}}$C in COS-7, COS-7, HeLa, HeLa, pre-BA, and BA cells, respectively [37,39,42,43,78]. A somewhat larger increase was detected with Eu-TTA-containing nanoparticles (NPs_ Eu2) in CHO-K1 cells (3–4 ${}^{{}^{\circ}}$C) [48]. The quantum-dots-based methodology allowed the determination of an average increase in T_cyto_ of ∼0.94 ${}^{{}^{\circ}}$C in SH-SY5Y cells, although the variations between individual dots were significant: from −2.5 to +5.5 ${}^{{}^{\circ}}$C [24]. By contrast, tsGFP1 and Eu-TTA/Rhod 101 nanosheets registered no increase in T_cell_(T_cyto_) in HeLa, BAT, and cardiomyocytes [29,66]. A Au/Pd thermocouple immersed into the neuronal cells detected a very short-lived (one second) temperature spike of 7.5 ${}^{{}^{\circ}}$C in response to the addition of the “mild” uncoupler BAM15 [83]. However, the nature of the observed spike remains an open question, since the effects of BAM15 on membrane potential and heat production did not coincide in time. A parallel measurement of the extracellular temperature revealed only a minimal increase of +0.1–0.2 ${}^{{}^{\circ}}$C. The upper limit of the uncoupler-dependent rise in T_cyto_ was observed with an FPT [80], a linear cationic analogue of an FPT from [39]. Using a cationic FPT, the authors determined a T_cyto_ elevation of 10 ${}^{{}^{\circ}}$C in HEK293T cells [80].

The results presented in Table 1 reveal at least one obvious discrepancy. In fact, the measurements with anionic and cationic FPTs, which differ by the presence of either SPA or APTMA leading groups in the structure, gave an order of magnitude difference in the values of the FCCP-stimulated T_cyto_ elevation in kidney cells [39,80]. Further, the positively charged sensor AP4-FPT, which bears the more photostable fluorescent moiety DBThD-AA [37] instead of DBD-AA in an FPT [80], reported a sixfold lesser FCCP-dependent increase in T_cyto_ than the latter. In addition, it was reported that an anionic FPT measured a rise in T_cyto_ of up to 4 ${}^{{}^{\circ}}$C in the neighborhood of mitochondria [39]. Thus, one can conclude that both photobleaching and mitochondria-targeting sequences may affect the accuracy of temperature determination in the cytosol by nanogel thermometers.

The question arises as to which of the values obtained are physiologically relevant? Is it possible to correlate the observed increase in T_cyto_ with the rate of heat generation in the cell and mitochondria during mitochondrial uncoupling? As is discussed in Section 2.3.4 Mitochondria, the rate of heat production in the mitochondria of intact cells in the absence of an uncoupler (“resting state”) can be up to 50% that of the rate when respiration and phosphorylation are completely uncoupled. Therefore, the relatively high “resting-state” T_cyto_ and its relatively low FCCP-dependent elevation detected in SH-SY5Y and BA cells [24,78] may reflect the real rates of the production of warmth in mitochondria and the actual local temperature in their neighborhood. However, it is not possible to make an unambiguous conclusion whether the given figures reproduce the real temperature in the cytosol/its compartments or are calibration errors. It can be assumed that a good experimental protocol, including controls with the maximum and minimum possible cellular heat production, would reveal many methodical errors associated with signal calibration, interfering interactions, and intracellular localization of sensors that are not obvious from a physical point of view.

The uncoupling of mitochondria from BAs via the β-ARs-dependent UCP1 activation [17,18,35,78,82] or by the overexpression of UCP2 in HEK293T cells [81] led to an increase in T_cyto_ by 1.25–1.4 ${}^{{}^{\circ}}$C (1 and R-CFPT) [78,82], and outside the cells (microcantilever, diode thermal sensor) and in the cell suspension (thermocatheter) by 0.1–0.22 [17,18] and 0.16 ${}^{{}^{\circ}}$C [81], respectively. The temperature of the lipid droplets (DTG) in BAs remained almost constant (−0.24 ± 1.0 ${}^{{}^{\circ}}$C) [35]. In addition, 1 showed an excellent correlation of a CL316.243-dependent T_cyto_ increase with the UCP1 protein level in mitochondria [82]. These data mutually agree in the frame of the model where intracellular compartments and the cell surface are separated from each other by membrane structures and lipid droplets (Figure 1b).

Before considering the thermal effects of modulators of the level of cytosolic Ca^2+^, it should be noted that the entry of Ca^2+^ through the plasma membrane or its release from the intracellular depot cannot be considered exclusively in terms of regarding the activation of SERCA or myosin ATPase [34,66,67]. Indeed, the entrance of Ca^2+^ into the cytosol causes the immediate activation of three main heat-producing processes: ATP-dependent Ca^2+^ pumping into the ER/SR, ΔΨ_m_-dependent MCU-mediated Ca^2+^ accumulation by mitochondria, and actin–myosin ATP-dependent contraction in muscle cells [71,84,85,86,87,88,89,90,91] (Figure 2 and Figure 3). In mitochondria, Ca^2+^ causes a transient decrease in the mitochondrial transmembrane potential (ΔΨ_m_), the activation of respiration, and the suppression of the ATP synthesis [71,85,92]. A prolonged elevation in cytosolic Ca^2+^ can evoke irreversible mitochondrial damage, the induction of the so-called permeability transition pore (PTP), which prevents the ATP synthesis and, as a consequence, the ATPase activity of SERCA [89,90,91,93]. In addition, PTP opening causes the mitochondrial uncoupling and activation of the hydrolysis of cytosolic ATP [90]. The ATPase activity of the actin–myosin complex may be suppressed even earlier due to its constant Ca^2+^-dependent contraction (Figure 2) [94,95,96]. Thus, a prolonged activation of the Ca^2+^ influx in the cytosol should consequently activate several heat-generating systems, whose contribution to total heat production will depend on the cell type and incubation conditions. The elucidation of the contribution of each system to the total increase in T_cyto_ or in the temperature of different organelles will require the development of protocols capable of separating the processes of warmth production.

The activation of SERCA in HeLa cells by the Ca^2+^-ionophore ionomycin gave somewhat contradictory results upon the measurements of T_cyto_ by Eu-TTA-based sensors. Nanosheets revealed no change [66], micropipettes detected only a minor (≤1 ${}^{{}^{\circ}}$C) but thapsigargin-sensitive increase [49], while nanoparticles registered a pronounced T_cyto_ rise (+2 ${}^{{}^{\circ}}$C ranging from +0.5 to +3.5 ${}^{{}^{\circ}}$C) [77]. However, the latter was very transient (100–150 s) with the gradual return to the initial temperature (T_0_). (The authors stated that the concentration of Ca^2+^ was sufficient to kill cells fast.) Nanoparticles were distributed randomly in the cytosol: near mitochondria, ER, and plasma membrane, which apparently is a reason for the broad variation in experimental values. (Similar results were obtained with quantum dots in NIH/3T3 cells (+1.8 ${}^{{}^{\circ}}$C), again with a broad temperature distribution from −2 to +8 ${}^{{}^{\circ}}$C [26]. Low Ca^2+^ concentration in the medium was associated with a longer elevation in temperature (>5 min). Remarkably, quantum dots were predominantly colocalized with mitochondria.) In addition, nanosheets were unable to recognize a difference higher than 0.1 ${}^{{}^{\circ}}$C between the temperature of the medium and HEK293T cells bearing the mutated type 1 ryanodine receptor, which supports a constant Ca^2+^ leak and causes malignant hyperthermia in muscles [66]. In cardiomyocytes, myotubes, and hippocampal neurons, nanosheets and the fluorescent protein mCherry were unable to detect the temperature rise caused by the ryanodine receptor agonist caffeine and electrical stimulation [66,67].

Thus, this experimental protocol does not allow one to draw an unambiguous conclusion about the system responsible for the heat production upon the activation of Ca^2+^ influx into the cytosol. Correct controls that exclude all players but one could improve the situation considerably. The application of thapsigargin could be a good control of the contribution of SERCA to local heart production. However, thapsigargin added one hour before ionomycin [49] cannot be considered as an appropriate control since it causes a rise in the concentration of cytosolic Ca^2+^, which can irreversibly damage mitochondria and suppress their thermogenic ability (Figure 2). This is also true for the actin–myosin ATPase activity in myocytes. In any case, the transient ionomycin-induced elevation in local temperature seems to be insufficient for a detectable change in the temperature of a whole cell.

Taking into account the uncertain nature and localization of heat-generating processes, as well as the errors and limitations of measurement methods, it is clear that the study of the effect of various damaging agents (Aβ42, NaCN, inhibitors of DNA topoisomerase I and II) on T_cell_(T_cyto_) demands even more attention to the experimental protocols and necessary controls [15,80,81]. Moreover, these agents are hardly suitable for the validation of methods and the calibration of thermal gradients since they can indirectly affect many intracellular processes making the prediction and evaluation of the correctness of the results impossible.

#### 2.3.2. ER and SR

Table 2 shows the results of temperature measurements in the ER/SR using several fluorescent ER-targeted probes. Most of the probes are BODIPY derivatives: ER thermo yellow [34,66,67], ER thermo green [35], and ERtherm-AC [59] (Table 2). In addition, the data for the fluorescent ER-targeted protein construct tsGFP1-ER are also provided [29]. The temperature in the ER/SR (T_ER_) was modulated by two groups of agents: direct and indirect mitochondrial uncouplers (FCCP, isoproterenol, forskolin) and modulators of cytosolic Ca^2+^, namely ionomycin, caffeine, thapsigargin, and cyclopiazonic acid (the latter two are SERCA inhibitors). From the data in the table, it follows that the measurements of T_ER_ using tsGFP1 and ERtherm-AC may require special attention to the calibration of fluorescence signals in the selected cell types [29,59]. Indeed, according to the calibration curves presented, the values of changes in fluorescence correspond to physiologically and physically inexplicable temperature alterations [29,59] and do not correlate with the results obtained with the use of other sensors [34,35,66,67]. Further, FCCP presumably interferes with the temperature measurement using ER thermo yellow [66]. The treatment of HeLa cells with ionomycin caused a rise in T_ER_ of ∼1 ${}^{{}^{\circ}}$C [67], 1.7 ${}^{{}^{\circ}}$C [34], and ∼3 ${}^{{}^{\circ}}$C [66] lasting for more than 150, for 200–250, and more than 500 s, respectively. Similarly, caffeine induced a fast and transient (∼100 s) elevation in cytosolic Ca^2+^, which was followed by a 1.6 ${}^{{}^{\circ}}$C increase in T_ER_ in myotubes [67], while the thapsigargin-related Ca^2+^ elevation was accompanied by a ∼0.5 ${}^{{}^{\circ}}$C temperature rise. These data strongly suggest that SERCA-dependent heat production may be responsible for an elevation of about 1 ${}^{{}^{\circ}}$C in T_ER_ in myotubes. The remaining increase can be attributed to actin–myosin ATPase and transiently uncoupled mitochondria (Figure 2). Isoproterenol induced a T_ER_ rise of ∼0.6–0.7 ${}^{{}^{\circ}}$C [35], which supports the role of mitochondria in ER heating and is consistent with the results obtained on myotubes, corrected for cell type.

#### 2.3.3. Nuclei

The phenomenon of the increased temperature in the nuclei (and centrosomes) in comparison with the neighboring cytosol was described in several works [28,37,39,40]. The processes of transcription and DNA replication occurring in the cell nucleus are accompanied by a high energy consumption (in the form of ATP), which should lead to a significant yield of warmth [73,74]. Alternatively, the external membrane of the nuclear envelope contains multiple copies of SERCA capable of ATP-dependent Ca^2+^ pumping into the lumen [70,71,75]. However, the nucleus itself occupies a significant volume in a cell, which may make the specific heat production quite low. Therefore, the question arises whether the processes occurring in the nucleus can cause its local heating. Table 3 shows the results of studies that compared temperatures in the nucleus (T_Nuc_) and the surrounding cytoplasm (T_Cyto_). Fluorescent polymer probes (1, FPT, AP4-FPT) [37,39,40], a low-molecular-weight BODIPY derivative NTG [35], and a ratiometric couple of fluorescent proteins gTEMP [28] were used as sensors. Using polymeric probes, T_Nuc_ was found to be 0.5 ${}^{{}^{\circ}}$C (HeLa), 1 ${}^{{}^{\circ}}$C (HeLa), 1 ${}^{{}^{\circ}}$C (COS-7), and 1 ${}^{{}^{\circ}}$C higher (HEK293T) than T_Cyto_ [37,39,40]. In addition, it was shown that T_Nuc_ depends on the cell cycle phase, being higher in the G1 phase (+0.7 ${}^{{}^{\circ}}$C) and lower in the S/G2 phase (−0.03 ${}^{{}^{\circ}}$C) (COS-7) [39]. It is noteworthy that gTEMP, possessing the highest molecular weight among nuclear thermal sensors, determined the highest temperature gradient between nuclei and cytosol (+3 ${}^{{}^{\circ}}$C) [28]. In contrast, a low-molecular-weight NTG sensor, which bears the DNA-targeted Hoechst 33258 moiety, found equal values of T_Nuc_ and T_Cyto_ in BAs [35]. Presumably, this phenomenon demands further elucidation and experimental examination. Indeed, if the processes of replication and transcription are responsible for nuclear heating, then one could expect the maximum nuclear temperature during the S phase, when DNA replication occurs (Figure 2a). However, this is not true [39]. Moreover, the energy spent in the process of DNA replication in the cell nucleus is only a small fraction of the energy spent on nucleotide biosynthesis, which occurs in the cytosol [73,74]. Alternatively, if heat generation is due to ATP-dependent Ca^2+^ pumping into the lumen of the nuclear envelope, then the nuclear surface should be warmer than the nucleoplasm (Figure 2a). In addition, low-molecular-weight sensors (especially DNA-targeted) should report a lower thermal gradient between nuclei and cytosol than large proteins, whose passage through the nuclear pores is restricted. This is consistent with experimental data [28,35]. However, whether the decrease in fluorescence of high-molecular-weight sensors, which corresponds to the increased temperature, is associated with the restriction of their passage into the nucleus at certain phases of the cell cycle, with interaction with DNA or histones [97], or actually reflects an increase in T_Nuc_ is currently obscure. Thus, the question of a “warm nucleus” requires further exploration.

#### 2.3.4. Mitochondria

One of the main functions of mitochondria in a cell is to maintain the pool of adenine nucleotides in a reduced state. The efficiency of the storing energy in the form of ATP is about 50% (between 40% and 60%), with the rest being dissipated as heat [68]. However, in mitochondria where the respiration and phosphorylation are uncoupled by a protonophore, the energy produced in the course of substrate oxidation is totally dissipated (Figure 3). The permanent phosphorylation of ADP produced by numerous ATP consumers keeps the mitochondria of “resting” cells in the “quasi 3d” state of respiration (intermediate state between third and fourth state) according to Chance, since the cytosolic ADP/ATP ratio, being always very low, cannot be equal to zero [70,98].

Therefore, mitochondria are considered to be one of the main sources of heat in a cell. The analysis of numerous studies in which cellular respiration was measured with SeaHorse or Oroboros and other Clarck-type oxygen sensors shows that the inhibitor of F-ATP synthase oligomycin causes a suppression of respiration, which is comparable with, or even greater than the activation of respiration caused by the addition of an uncoupler to the “resting” cells. For instance, in the human MDA-MB-468 cells, oligomycin suppressed the “resting state” respiration approximately by 80%, and KCN caused an additional ∼10% suppression, while FCCP accelerated respiration by approximately 50% [99]. If we take these values as some standard, then mitochondrial heat production in the presence of oligomycin and respiratory chain inhibitors will tend to zero. Proton leaks responsible for respiration in the presence of oligomycin (Figure 3a) and oxidative phosphorylation will give about 5–10% and 40% (together 50%) of total uncoupler-releasable warmth, respectively. Such a high rate of heat production in mitochondria of “resting” cells raises the question of how hot mitochondria can be in comparison with cytosol and other organelles in physiological and pathological states. Taking into account that mitochondria play a crucial role not only in the energetic metabolism of cells but also in the control of cell death, the clarification of this issue is of great importance for a better understanding of the regulation of these processes.

Maximum heat production by mitochondria is limited by their maximum respiratory capacity, which, in turn, is determined by the number of respiratory complexes per cristae and the crista density [100,101]. The maximum respiratory capacity is highly (several-fold) variable in mitochondria of the same tissues from different biological species [87,102,103], different tissues of the same species [103], or the same tissues and species at different metabolic demands [104]. (Moreover, the respiratory capacity of different cristae of the same mitochondrion may differ considerably [105].) The mitochondria of BAs, for instance, have a higher content of ETC complexes and crista density and thus higher respiratory and heat-producing capacities than epithelial cells [106]. Since in the majority of studies, the maximum respiratory capacity of cells was not specifically studied, the data in Table 4 (as well as in other tables) are grouped by cell types.

To date, several types of sensors have been developed to measure the mitochondrial temperature (T_mito_) in intact cells (Table 4): mitochondria-targeted fluorescent proteins (tsGFP1-mito, gTEMP, emGFP-Mito) [28,30,107], fluorescence polymers [108], and low-molecular-weight sensors comprising rhodamine- (Mito-RTP, Mito-TEM, Mito-TEM2 [33,58,109,110], rhosamine-(MTY) [32,111], or/and BODIPY-based reporters (MTG, Mito-TEM2) [33,35]. The selective delivery of protein sensors to mitochondria is achieved by introducing mitochondria-leading sequences into the protein structure, e.g., leader sequences of the E1 α-pyruvate dehydrogenase complex [30]. The delivery of polymeric and low-molecular-weight sensors into mitochondria is based on the ΔΨ_m_-dependent accumulation of cations in the negatively charged mitochondrial matrix. Rhodamine- and rosamine-related sensors are cations at normal cellular pH, while the BODIPY-based sensors are usually supplemented with the TPP^+^ groups. The weakness of this approach is a high dependence of the concentration (and hence fluorescence) of the label in the mitochondria on the changes in ΔΨ_m_ (Figure 3b), which, in turn, is one of the most important regulators of the rate of respiration and heat production [112,113,114]. Therefore, the use of such sensors, being much simpler technically than the induction of the expression of mitochondria-targeted fluorescent proteins, requires special attention to experimental protocols and the calibration of the temperature response.

Table 4 shows the results of measurements of T_mito_ using various types of sensors. The modulators of the rate of heat production were direct (CCCP/FCCP) [28,29,30,50,58,109,110,111] and indirect uncouplers (norepinephrine, isoproterenol) [29,35,58], inhibitors of the respiratory chain (rotenone, antimycin A, KCN) [29,32] and oxidative phosphorylation (oligomycin) [32], and different activating and damaging agents with diverse and multiple targets in cells *S. aureus*, lipopolysaccharide, and phorbol ether [33,110] (see Figure 3a,c). The results obtained using mitochondria-targeted fluorescent proteins show that the activation of respiration by direct uncouplers causes an increase in T_mito_ of 3–5 ${}^{{}^{\circ}}$C (HeLa) [30], 4–6 ${}^{{}^{\circ}}$C (HeLa), 5–6 ${}^{{}^{\circ}}$C (BA) [29], and 6–9 ${}^{{}^{\circ}}$C (HeLa) [28]. tsGFP1-mito [29] and emGFP-Mito [30] demonstrated a somewhat clearer mitochondrial localization than gTEMP [30]. However, in all cases, a considerable T_mito_ heterogeneity was observed, which reached about 20 (25–45 ${}^{{}^{\circ}}$C) without stimulus [29] and even 40 ${}^{{}^{\circ}}$C (15–57 ${}^{{}^{\circ}}$C) in the presence of FCCP [30]. Unfortunately, in two studies out of three, the experimental protocol did not allow one to assess the mitochondrial temperature before the uncoupler addition [28,30]. In a single study, the application of rotenone demonstrated that T_mito_ could be diminished by 10–12 ${}^{{}^{\circ}}$C below T_med_ (our estimation based on the provided calibration curves) [29]. A simultaneous treatment with CCCP and rotenone caused a decrease of approximately 2 ${}^{{}^{\circ}}$C in T_mito_. This, as well as the fact that mitochondria possessing ΔΨ_m_ were determined to be hotter than others clearly indicates that a proper calibration of the thermal response of fluorescent proteins requires the employment of a set of modulators of the mitochondrial functional state. Ideally, these modulators should allow the mitochondria in a cell to be stepwise-transferred from the state with the highest possible heat production to the state with the least one at different ΔΨ_m_.

**Table 4 ijms-24-16955-t004:** Stimulus-dependent changes in the mitochondrial temperature.

Cell Type *	Assay	Thermosensor **	Stimulus ***	$T_{\mathrm{med}}\left(T_{0}\right)/\Delta T$ ****	Ref.
HeLa	Confocal fluorescence microscopy	MTY, mito thermo yellow	5 μM FCCP	Dye release (−0.4F within 4 min) matching to ΔT=+25 ${}^{{}^{\circ}}$C (our estimation)	[111]
HeLa	Confocal fluorescence microscopy	Mito-TEM 2, rhodamine B-BODIPY construct bearing formaldehyde anchor	No stimulus 10 μL/mL *S. aureus* (60 min) 20 μg/mL LPS (30 min)	$T_{med}=37$ ${}^{{}^{\circ}}$C (heterogeneous $T_{0}$ = 35–40 ${}^{{}^{\circ}}$C) /+6 ${}^{{}^{\circ}}$C (Heterogeneous $T_{st}$ = 40–46 ${}^{{}^{\circ}}$C) /+6 ${}^{{}^{\circ}}$C (heterogeneous $T_{st}$ = 40–46 ${}^{{}^{\circ}}$C)	[33]
HeLa	Fluorescence microscopy	Mito-RTP, mitochondrial ratiometric temperature probe, rhodamine B-CS NIR dye construct	10 μM FCCP	$T_{med}=34$ ${}^{{}^{\circ}}$C/+3 ± 1 ${}^{{}^{\circ}}$C (our estimation)	[109]
HeLa	Fluorescence intensity, microplate reader	T sensing probe, polyNIPAm-VBC-DACC-CTPP	200 μM FCCP removal of FCCP	$T_{med}=32.6$ ${}^{{}^{\circ}}$C/+2.4 ${}^{{}^{\circ}}$C ;$T_{0}=34.5$ ${}^{{}^{\circ}}$C/−2 ${}^{{}^{\circ}}$C (our estimation)	[108]
HeLa	Confocal fluorescence microscopy, peak fraction analysis	emGFP-Mito, mitochondria-targeted GFP (CellLight BacMam 2.0)	10 μM FCCP	$T_{med}=37$ ${}^{{}^{\circ}}$C/+3–5 ${}^{{}^{\circ}}$C (duration of maximum rise ∼300 s) organelle thermal heterogeneity (from 15 ${}^{{}^{\circ}}$C to 57 ${}^{{}^{\circ}}$C).	[30]
HeLa	Fluorescence microscopy	gTEMP, coupled fluorescent proteins	10 μM FCCP	$T_{med}=37$ ${}^{{}^{\circ}}$C/+6–9 ${}^{{}^{\circ}}$C organelle thermal heterogeneity.	[28]
HeLa, BA	Confocal fluorescence microscopy	tsGFP1-mito, mitochondria-targeted GFP-TlpA construct	No stimulus 10 μM rothenone 10 μM CCCP CCCP+ rothenone 10 μM CCCP 10 μM NE	Organelle thermal heterogeneity (25–45 ${}^{{}^{\circ}}$C) $T_{med}=37$ ${}^{{}^{\circ}}$C/−10 ${}^{{}^{\circ}}$C (HeLa, our estimation) /+4–6 ${}^{{}^{\circ}}$C (HeLa, our estimation) /−2 ${}^{{}^{\circ}}$C (HeLa, our estimation) $T_{med}=37$ ${}^{{}^{\circ}}$C/+5–6 ${}^{{}^{\circ}}$C (BA) /+3–5 ${}^{{}^{\circ}}$C (BA)	[29]
BA	TCSPC system-based FLIM	MTG, mitochondria thermo green, BODIPY-TPP+ fusion	1 mM ISO (50 min)	$T_{med}=37$ ${}^{{}^{\circ}}$C/+2.8 ± 2.7 ${}^{{}^{\circ}}$C	[35]
BA	Confocal fluorescence microscopy	Ratiometric rhodamine B/rhodamine 800 pair	10 μM CCCP (30 min) 100 nM NE	T_med_ = 33 ${}^{{}^{\circ}}$C/+15 ${}^{{}^{\circ}}$C /+4 ${}^{{}^{\circ}}$C (Our estimation)	[58]
MCF-7	Confocal fluorescence microscopy	Mito-TEM, rhodamine B bearing targeting and anchoring moieties	50 μM PMA (30 min) 50 μM CCCP (30 min)	$T_{med}=29$ ${}^{{}^{\circ}}$C/+3 ${}^{{}^{\circ}}$C $T_{med}=37$ ${}^{{}^{\circ}}$C/(brighter fluorescence)	[110]
HEK 293, primary skin fibroblasts	Fluorescence	MTY, mito thermo yellow	No stimulus Anoxia 0.8 mM KCN 3 μM rotenone 1 μM antimycin A 5 μM oligomycin	$T_{0}=50$ ${}^{{}^{\circ}}$C /−10.5 ${}^{{}^{\circ}}$C /∼−10.5 ${}^{{}^{\circ}}$C /∼−10.5 ${}^{{}^{\circ}}$C /∼−10.5 ${}^{{}^{\circ}}$C /∼−9 ${}^{{}^{\circ}}$C	[32]

Notes: *—MCF-7, human breast carcinoma. **—AEMA, 2—aminoethyl methacryate; BMA, butyl methacrylate; CTPP, 4-Carboxybutyl)triphenylphosphonium bromide; DACC, 7-(diethylamino) coumarin-3-carbaldehyde; EGDMA, ethyleneglycol dimethacrylate; emGFP-Mito, GFP fused to the leader sequence of E1 a-pyruvate dehydrogenase complex (CellLight BacMam 2.0); L2, 4-(4,6-di(1H-pyrazol-1-yl)-1,3,5-triazin-2-yl)-N,N-diethylaniline; Mito-TEM 2, aminated rhodamine B linked to formaldehyde BODIPY derivative by benzaldehyde moiety; Mito-RTP, rhodamine B and CS NIR dye conjugated by hexamethylenediamine linker; mito thermo yellow, N,Ndibutyl-6-imino-9-(3-piperidin-1-ylphenyl)xanthen-3-amine; MMA, methyl methacrylate; VBC, 4-vinylbenzylchloride. ***—LPS, lipopolysaccharide; PMA, Phorbol-12-myristate-13-acetate. ****—$T_{med}$, $T_{0}$, and $T_{st}$, the temperature of the medium, mitochondria before stimulus, and mitochondria after stimulus, respectively; $\Delta T=T_{st}-T_{0}$. ***—$T_{med}$, $T_{cyto}$, $T_{nuc}$, and $T_{centro}$, temperature of the medium, cytosol, nuclei, and centrosomes, respectively; $\Delta T_{nuc}=T_{nuc}-T_{cyto}$; $\Delta T_{centro}=T_{centro}-T_{cyto}$.

A fluorescent cationic-polymer sensor detected a relatively small (∼2.5 ${}^{{}^{\circ}}$C) uncoupler-dependent rise in T_mito_ (HeLa) [50]. The subsequent removal of FCCP from the medium caused the fast return of the temperature to near-initial values. However, the reliability and biological validity of these results are questionable due to the fact that an extremely high concentration of the uncoupler (200 μM) was used, which could inhibit the activity of the respiratory chain complexes [115]. In addition, it is hardly possible to remove such a large amount of the uncoupler from the cell.

Measurements with an MTY sensor showed, on the one hand, an extremely high “resting-state” T_mito_ (∼50 ${}^{{}^{\circ}}$C) [32] and an extremely high sensitivity of the fluorescent signal to ΔΨ_m_ changes on the other [111]. The dissipation of ΔΨ_m_ by an uncoupler caused the immediate release of the dye and a decrease in the fluorescence, which would correspond to an elevation of approximately 25 ${}^{{}^{\circ}}$C in T_mito_ within a few minutes [111].

It should be mentioned that in several cancer cell lines, in contrast to skin fibroblasts, MTY was found to become ΔΨ_m_-independent after the accumulation by mitochondria [116]. The mechanism underlying this phenomenon is not entirely clear. The authors believe that this may be connected with the covalent binding of MTY to the matrix proteins. Another possible reason is the effect of pH on the ΔΨ_m_ dependence of MTY distribution. Indeed, the amide group of MTY should be protonated by 85% (calculation using ACD/Labs software) at normal cytosolic pH (∼7.2) and predominantly deprotonated at the normal pH of the mitochondrial matrix (7.8–8.0), thus losing the ΔΨ_m_ dependence. Hence, H^+^ buffering with the P_i_/adenine nucleotide system and the modulation of this buffering by P_i_ transport and ATP synthesis/hydrolysis should affect the matrix pH and ΔΨ_m_ dependence of MTY. Therefore, the application of sensors with a high ΔΨ_m_ or pH sensitivity makes the calibration of the mitochondrial thermal response quite risky.

Nevertheless, the effect of respiratory chain inhibitors on the mitochondrial temperature (measured decrease of ∼10 ${}^{{}^{\circ}}$C (7–12 ${}^{{}^{\circ}}$C)) implies that the organelles in the “resting state” may be to some extent warmer than the surrounding cytosol [32].

Rhodamine-based sensors, which are also ΔΨ_m_-sensitive, indicated an uncoupler-dependent elevation in T_mito_ by ∼3 (Mito-RTP) [111] and 15 ${}^{{}^{\circ}}$C (Rhod B/Rhod 800) [58] (our estimation in both cases) and a norepinephrine-dependent elevation of 4 ${}^{{}^{\circ}}$C (Rhod B/Rhod 800) [58]. Moreover, the activators of various intracellular processes, such as *S. aureus* (MOI value was not provided), lipopolysaccharide [33], and phorbol ether [110] elevated Tmito by ∼6, 6, and 3 ${}^{{}^{\circ}}$C, respectively. This approach avoids the use of strong ΔΨ_m_ modulators but does not allow one to assess the correctness and biological validity of the results. In particular, it was previously reported that lipopolysaccharide caused the accumulation of ATP and an increase in ΔΨ_m_ in HeLa cells [117] and that phorbol ester caused a G1 phase arrest in MEF-7 cells [118], which should decrease the energy demands of cells. When measured with an MTG sensor, isoproterenol caused an increase in T_mito_ of ∼3 ${}^{{}^{\circ}}$C [35].

It is obvious that minimal mitochondrial heat production occurs when the respiratory chain and F-ATP synthase are blocked by specific inhibitors (Figure 3c). (The inhibition of ATP synthase is important for the prevention of heat generation upon the reversion of the reaction of ATP synthesis for ΔΨ_m_ restoration.) On the contrary, maximum heat production (about 440 kJ/mole of O_2_ consumed) takes place upon mitochondrial uncoupling by direct and indirect uncouplers and Ca^2+^ in the presence of substrates providing NADH to the complex I [69,70]. In both cases, ΔΨ_m_ is disrupted and a considerable part of ΔΨ_m_-dependent sensors exits from mitochondria. Presumably, measuring the fluorescence of the traces of ΔΨ_m_-dependent sensors in the presence of respiratory chain inhibitors and uncouplers is the only possibility to calibrate the range of the temperature fluctuations in mitochondria. Thus, one may conclude that two strategies for correct measurements of T_mito_ and its fluctuations are possible: the use of ideal environment-independent sensors and the use of ideal protocols including all necessary controls for the calibration of temperature response on the changes in the functional state of mitochondria.

#### 2.3.5. Physiologic and Pathophysiologic Thermal Limits for Mitochondria

The maximum reported values of mitochondrial heating (≥+15 ${}^{{}^{\circ}}$C) [58,59] and inhibitor-dependent cooling (ca. −10–−12 ${}^{{}^{\circ}}$C) [29,32] regardless of their correctness raise the question of how hot mitochondria can be in physiologic and pathophysiologic states without an irreversible loss of their functionality [29,62,63,119]. Cells are known to be very sensitive to temperature elevation. In myotubes, a local heating of ∼5 ${}^{{}^{\circ}}$C can provoke the Ca^2+^-independent contraction of sarcomeres [120]. An hourly increase in T_med_ to 47 ${}^{{}^{\circ}}$C leads to a 40–50% death of human metastatic breast adenocarcinoma MDA-MB-468 cells and adherent mouse fibroblast L929 BALB/c cells (bearing mitochondria from platelets of the mouse strain BALB/cJ) over the next 6 hours [119]. For the human renal carcinoma cell line Caki-2, approximately 10 min of heating to 47 ${}^{{}^{\circ}}$C is sufficient to almost complete the suppression of viability [121].

Recently, Chrétien and co-workers claimed that the mitochondria of HEK293 cells in a “fully functional” state, i.e., in the intermediate 3–4 state of respiration (according to Chance) are ∼10 ${}^{{}^{\circ}}$C (7–12 ${}^{{}^{\circ}}$C) hotter than the medium [32]. This was demonstrated using the potential-sensitive dye MTY, in anaerobic conditions, and at high concentrations of OXPHOS inhibitors. This allowed the authors to state that the normal operating temperature of mitochondria is close to 50 ${}^{{}^{\circ}}$C, which implies the possibility of long-term functioning of OXPHOS complexes (and mitochondria) at this temperature without loss of activity. As we noted in Section 2.3.4, the reported values of T_mito_ are not convincing because of the high sensitivity of MTY fluorescence to changes in ΔΨ_m_. Since complexes I and III are characterized by a high flux control coefficient [60,122], the inhibition of these complexes (e.g., by rotenone and antimycin A) should inevitably lead to the ΔΨ_m_ dissipation and MTY release. In living cells, these will be delayed for several minutes due to the temporary support of ΔΨ_m_ by F-ATP synthase (see Figure 3b), which makes the calibration of the temperature response extremely complex.

Interestingly, in the same work, the authors showed that when HEK293 cells, primary skin fibroblasts, and isolated mitochondria were incubated at elevated temperatures (from 38 to 58 ${}^{{}^{\circ}}$C), different respiratory complexes showed different temperature optima: for complexes II-III, III, and IV, the thermal optimum was ∼50 ${}^{{}^{\circ}}$C, 46 ${}^{{}^{\circ}}$C for complex V, and 38 ${}^{{}^{\circ}}$C for complex I with a dramatic inhibition at 42 ${}^{{}^{\circ}}$C. Unfortunately, we were unable to understand from the description how long the mitochondria and cells were exposed to high temperature (probably for 10 min), and whether the contribution of the mitochondria’s own temperature to the effect of heating was taken into account. Therefore, it is unclear how long respiratory complexes can resist a high temperature and whether the inhibition of complex I by a high temperature is reversible and condition-specific. It was previously shown that at high temperatures, a reversible dissociation of flavin mononucleotide and exposure of reactive thiols occur, which leads to the reversible deactivation of the complex [123,124].

In a more recent work, Moreno-Loshuertos and co-workers showed that a one-hour incubation of MDA-MB-468 and L929^Balbc^ cells and a 30–60-min incubation of hepatocytes and isolated liver mitochondria from C57BL/6J mice at a temperature higher than 43–45 ${}^{{}^{\circ}}$C caused the temperature- and time-dependent inactivation of complex I/complex I-containing supercomplexes and the degradation of different mitochondrial complexes and supercomplexes [119]. At submaximal temperatures, the respiratory chain inhibitor KCN protected complexes and supercomplexes in cells but strengthened the harm from heating in isolated mitochondria. The authors concluded that physiologic limits for the sustainable operation of mitochondrial (super)complexes are below 43 ${}^{{}^{\circ}}$C.

These results seem convincing because the authors performed their experiments both on cells and isolated mitochondria and provided comprehensive information about the experimental conditions. However, the conclusion on the thermal limit at 43 ${}^{{}^{\circ}}$C can hardly be final. First, the oxygen level in cultured cells is significantly higher than in mammalian tissues around mitochondria in vivo (∼218 vs. 1.5–15 μM at 37 ${}^{{}^{\circ}}$C) [125,126]. Thus, the toxic effect of oxygen on mitochondrial supercomplexes in mammalian tissues can be significantly lower than in cell culture. Second, as it was stated above, cells are quite vulnerable to heating. This may be due to the hyperactivation of ionic channels by warming, which, in turn, may cause mitochondrial damage due to, for instance, Ca^2+^ overload [72,127]. Therefore, heating of cells may be more harmful for mitochondria than the heating of mitochondria themselves. Third, the heating of isolated respiring mitochondria may increase their own elevated temperature to unknown values, which may exceed the specified temperature limits during the experiment. Moreover, the heating of isolated mitochondria may increase their susceptibility to the opening of Ca^2+^- and inorganic phosphate-dependent PTP [128], which causes (super)complexes inactivation and degradation [129,130,131]. In contrast, the inhibition of the respiratory chain leads to a dissipation of ΔΨ_m_ and the inactivation of mitochondrial nicotinamide nucleotide transhydrogenase, which is required to maintain the reactive thiols of mitochondrial complexes in a reduced state through NADPH recovery [132,133,134]. This, in turn, may increase the mitochondrial sensitivity to heating via another mechanism.

In line with these suggestions, the measurements of T_mito_ in the aggregates of isolated mouse brain mitochondria by a diamond thermometer insensitive to nonthermal parameters showed that the uncoupler elevated the T_mito_ by 4–22 ${}^{{}^{\circ}}$C above the ambient temperature (T_med_) [23]. Nevertheless, the absolute maximum of T_mito_ was found to be about 45 ${}^{{}^{\circ}}$C, i.e., close to the limits determined by Moreno-Loshuertos and co-workers [119]. Thus, a definitive conclusion about the (patho-)physiological temperature limits for mitochondria requires the development of sensors or protocols that would circumvent all specified pitfalls.

#### 2.3.6. Avoidance of Thermal Damage

As has been already noted, heat production is ultimately determined by the rate of mitochondrial respiration [87]. The latter is regulated at many levels: the rate of delivery of gases and respiratory substrates [135], the transcription, translation, import and assembly of respiratory complexes [100], and the biogenesis and dynamics of cristae [101]. In simple cellular systems, this regulatory hierarchy is not complete, since the levels of oxygen and substrates are constant. If we accept the fact that mitochondria in a cell are really hotter than the surrounding cytosol, then a paradoxical situation arises: an increased temperature accelerates the biochemical reactions and enhances the fluidity [136] and proton permeability of membranes [137,138], which leads to the acceleration of respiration and heat production [87,137,139]. Therefore, there must be mechanisms that limit the mitochondrial heating and protect against its destructive consequences. One of these mechanisms may be the thermal inactivation of respiratory chain complexes when the so-called Arrhenius breakpoint temperature is exceeded [123,138,140,141], for example, due to the dissociation of noncovalently bound coenzymes [142].

Another mechanism of protection from overheating may be a change in the shape of mitochondria, the structure and volume of cristae, or a short-term opening of PTP. Thus, it was shown that the uncoupler of oxidative phosphorylation CCCP within 10 min dramatically changes the shape of mitochondria from filamentous to ring-shaped or globular with large cavities [143]. These cavities can be filled with the ER and cytosolic solution, which may facilitate the flow of heat away from mitochondria. However, the respiration inhibitors rotenone, antimycin A, and oligomycin caused changes in mitochondrial morphology similar to those caused by the uncoupler FCCP [144]. It has also been shown that the metabolic state of mitochondria (the rate of oxidative phosphorylation) can determine the structure and density of cristae. In particular, during hypoxia, the cristae expand, and when the respiratory substrates and oxygen are introduced, they narrow [145,146]. However, cristae were also reported to expand in the presence of ADP, i.e., with active breathing in state 3 [147]. Another study showed that both inhibitors (rotenone, antimycin A, oligomycin) and the uncoupler FCCP caused a decrease in crista density in mitochondria [144]. Thus, there is currently no convincing evidence that respiration-dependent changes in T_mito_ can affect the morphology of cristae and mitochondria.

At the same time, it is well known that an elevated temperature promotes the opening of the Ca^2+^-dependent PTP in mitochondria [130,148]. It was previously shown that the short-term opening of the PTP promotes the release of accumulated Ca^2+^ and reactive oxygen species, preventing irreversible damage to the organelles [149]. Perhaps the same mechanism exists to overcome critical thermal overloads.

The heating of cells leads to an increased expression of heat-shock proteins (HSPs) in mitochondria [150,151,152]. Theoretically, high levels of HSPs in mitochondria can be an indicator of elevated mitochondrial temperature. Indeed, there is evidence that the uncouplers of oxidative phosphorylation [153], cold, which induces the upregulation of UCP1 and uncoupling of mitochondria in BAs [154,155], and extensive physical loads [156] increase the level of mitochondrial HSPs. At the same time, an increase in the expression of HSPs can occur under various stressful states and influences: aging [157], hypoxia [158], and exposure to respiratory chain inhibitors [153]. Also, preliminary heat shock may have a protective effect on mitochondria in various pathological conditions [159,160]. All this allows one to consider HSPs as the elements of nonspecific protection against stress but not as indicators of mitochondrial temperature. At the same time, HSPs apparently play an important role in the avoidance of irreparable temperature damage by mitochondria.

## 3. Conclusions

The rate of biochemical reactions, as the rate of chemical reactions, strongly depends on temperature. Therefore, measuring local temperature in the cell’s cytosol, various organelles, and subcellular structures appears to be an important and promising task, whose solution will be beneficial for a better understanding of the mechanisms of various metabolic, signaling, and pathophysiological processes in cells and organs. To date, a large set of instruments have been developed to measure local temperature at the cell-organelle level. These instruments are characterized by different principles of thermal detection, sensitivity, inertness to biological objects, and other features. The main criteria for the proximity of a particular sensor to the “ideal” can be considered the physiological relevance of the results obtained with its help, and, to a lesser extent, the laboriousness of the temperature measurement process and the possibility of its automation and scaling.

Although none of the proposed sensors can be considered ideal, many sensors are very good tools for measuring temperature in selected cellular compartments and organelles, in some cell types, or under certain conditions. In particular, fluorescent nanosheets seem to be a perfect sensor for the determination of cell surface temperature due to the internal control of photobleaching (equal rates of photobleaching near and under a living cell). Targeted BODIPY-derivatives (thermo-greens) are promising thermosensors for multiple cell compartments (nuclei, ER, lipid droplets), maybe except mitochondria. ER thermo yellow generates predictable and physiologically relevant thermal responses in ER/SR. A good photostability, relative inertness, and diffused distribution make fluorescent proteins a good choice for the measurements of cytosolic temperature. However, their comparatively low sensitivity, probable steric restrictions and disruption of stoichiometry in ratiometric couples demand due attention to the calibration of the thermal response and the selection of the object and the process for investigation. Nanoparticles, nanodiamonds, and quantum dots, being inert to the biological environment and relatively stable (nanodiamonds are exceptionally stable) to photobleaching, may be attractive tools for some specific tasks. Nevertheless, a small number of sensor particles per cell and their unpredictable intracellular distribution require more experiments and a more careful and complex analysis of the results. In general, fluorescent polymers (a quite heterogeneous group) show results consistent with those expected. At the same time, the possibility of their interfering interactions with biological macromolecules (DNA, RNA, histones) and effectiveness at targeting “hot” organelles remain unclear. In any case, the development and implementation of good protocols for the calibration of sensor temperature responses and control of their specificity will definitely improve the accuracy of determining the local temperature and the source of heat generation. This will also prevent the appearance of nonsense temperature values.

Currently, it is difficult to name a good sensor for measuring temperature in mitochondria. Fluorescent proteins, whose distribution does not depend on ΔΨ_m_, give a very large spread of temperature values, sometimes beyond any acceptable limits. The use of ΔΨ_m_-dependent polymers and low-molecular-weight sensors requires a constant monitoring of their mitochondrial concentration and a great ingenuity in developing protocols for calibrating the thermal responses to stimuli. The situation can be improved by the development of a ΔΨ_m_-dependent probe with an anchor (MitoTracker-like) and the use of special protocols for the calibration of photobleaching and temperature responses. In addition, an interesting challenge for future research remains the determination of physiological temperature limits for potentially “hot” organelles such as mitochondria and to a lesser extent, the ER and nucleus.

## Figures and Tables

**Figure 1 ijms-24-16955-f001:**
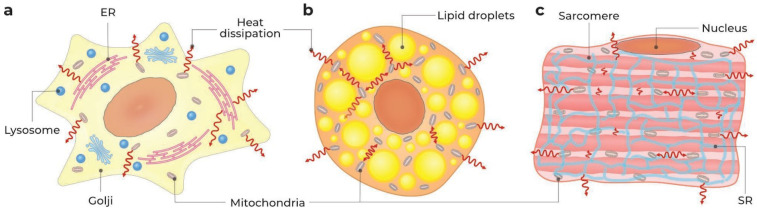
Morphology-based restrictions for heat dissipation in (**a**) undifferentiated cancer cell/fibroblast-like cell, (**b**) brown adipocyte, and (**c**) cardiomyocyte. Red arrows indicate the heat dissipation.

**Figure 2 ijms-24-16955-f002:**
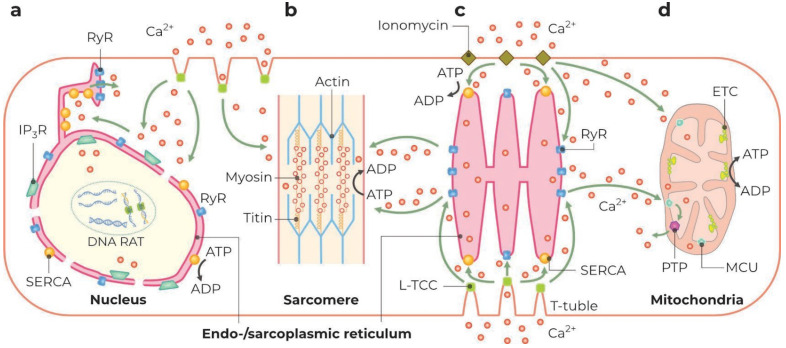
Intracellular Ca^2+^-responsive heat-generating targets. Cell compartments: nucleus (**a**), sarcomere (**b**), endo-/sarcoplasmic reticulum (**c**), and mitochondria (**d**). Green arrows show the Ca^2+^ propagation from extracellular space and intracellular depot; black arrows indicate the ATP–ADP conversion. Abbreviations: DNA RAT—DNA replication and transcription; ETC—electron transport chain; IP_3_R—inositol triphosphate receptor; L-TCC—L-type Ca^2+^ channel; MCU—mitochondrial calcium uniporter; PTP—permeability transition pore; RyR—ryanodine receptor; SERCA—Ca^2+^-ATPase of endoplasmic and sarcoplasmic reticulum.

**Figure 3 ijms-24-16955-f003:**
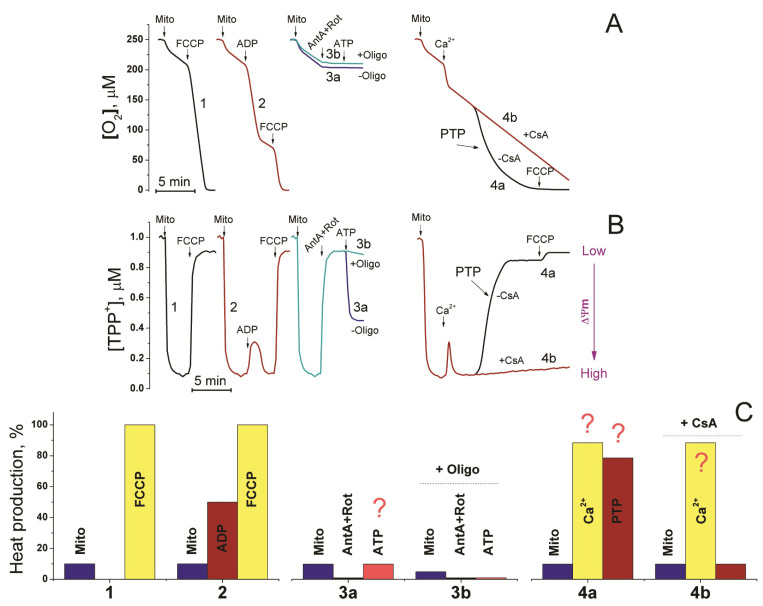
The dynamics of respiration rate (**A**) and ΔΨ_m_ (**B**) in different functional states of isolated mitochondria and the corresponding approximate rates of heat production (**C**). The illustrative idealized curves obtained with rat liver mitochondria (1 mg/mL) placed in the incubation medium (125 mM KCl, 20 mM sucrose, 10 mM HEPES (pH 7.3), 2 mM MgCl_2_, 2 mM KH_2_PO_4_, and 10 μM EGTA) and supplemented with 5 mM glutamate, 5 mM malate (**A**), and 1 μM tetraphenylphosphonium chloride (TPP^+^) (**B**). Arrows show the addition of mitochondria (Mito), uncoupler FCCP, ADP, respiratory chain inhibitors antimycin A and rotenone (AntA + Rot), ATP, and Ca^2+^ (to induce the permeability transition pore (PTP) opening). As indicated, the medium contained F-ATP synthase inhibitor oligomycin (+Oligo) and PTP inhibitor cyclosporine A (+CsA). Question marks indicate that exact values of heat production are not available and may be roughly deduced by the analogy with other processes.

**Table 2 ijms-24-16955-t002:** The stimulus-dependent changes in ER/SR temperature.

Cell Type	Assay	Thermosensor *	Stimulus	$T_{0}/\Delta T$ **	Ref.
HeLa	Fluorescent microscopy	ER thermo yellow	10 μM FCCP 2 μM ionomycin	Rotenone-insensitive nonthermal effects 36 ± 1 ${}^{{}^{\circ}}$C/+∼3 ${}^{{}^{\circ}}$C	[66]
HeLa	Fluorescent microscopy	ER thermo yellow	1 μM ionomycin	37 ${}^{{}^{\circ}}$C/+1.7 ± 0.4 ${}^{{}^{\circ}}$C (duration 200–250 s)	[34]
HeLa, C2C12 myoblasts, and differentiated myotubes	Time-domain FLIM, frequency-domain FLIM, confocal laser scanning microscopy	ER thermo yellow	2 μM ionomycin 1 mM caffeine 1 mM thapsigargin thapsigargin (30 min) + caffeine	37 ${}^{{}^{\circ}}$C/+0.93 ± 0.68 ${}^{{}^{\circ}}$C (HeLa) +1.6 ± 0.6 ${}^{{}^{\circ}}$C (duration > 150 s; myotubes) ΔT→0 (myoblasts) +0.38 ± 0.34 ${}^{{}^{\circ}}$C +0.41 ± 0.48 ${}^{{}^{\circ}}$C	[67]
C2C12 myoblasts and differentiated myotubes	Confocal microscopy	tsGFP1-ER, ER-targeted GFP-TlpA fusion protein	50 μM cyclopiazonic acid (SERCA inhibitor)	37 ${}^{{}^{\circ}}$C/−12 ${}^{{}^{\circ}}$C (our estimation, +0.1R, myotubes) ΔT→0 (myoblasts)	[29]
BA	TCSPC system-based FLIM	ETG, ER Thermo Green, unsymmetrical BODIPY derivative	1 mM ISO (50 min)	37 ${}^{{}^{\circ}}$C/+0.64 ± 1.8 ${}^{{}^{\circ}}$C	[35]
Mouse WT-1 pre-BA, human BA	Confocal fluorescence microscopy	ERtherm-AC, ER thermo yellow acetyl derivative	10 μM ISO 10 μM FCCP 10 μM forskolin	25 ${}^{{}^{\circ}}$C/+19.8 ${}^{{}^{\circ}}$C (our estimation, −0.5F) (pre-BA) 25 ${}^{{}^{\circ}}$C/+17.5 ${}^{{}^{\circ}}$C (our estimation, −0.4F) (BA)	[59]

Notes: *—ER thermo yellow, (*E*)-2-chloro-*N*-(4-(5,5-difluoro-3-(2-hydroxy-5-(trifluoromethoxy)styryl)-1-methyl-5H-4λ4,5λ4-dipyrrolo[1,2-c:2${}^{\prime }$,1${}^{\prime }$-f][1,3,2]diazaborinin-10-yl)phenyl)acetamide; BODIPY,4,4-di-fluoro-4-bora-3a,4a-diaza-s-indacene; **—$T_{0}$ was taken to be equal to $T_{med}$; ΔT is a stimulus-dependent increase/decrease in $T_{ER}$ in relation to $T_{0}$.

**Table 3 ijms-24-16955-t003:** The temperature gradient between nuclei, centrosomes, and cytosol.

Cell Type *	Assay	Thermosensor **	Stimulus	$T_{\mathrm{med}}\left(T_{\mathrm{cyto}}\right)$, $\Delta T_{\mathrm{nuc}}\left(\Delta T_{\mathrm{centro}}\right)$ ***	Ref.
HeLa	Fluorescent microscopy	gTEMP, coupled fluorescent proteins	No stimulus	$T_{med}=37$ ${}^{{}^{\circ}}$C , $\Delta T_{nuc}=+2.9\pm 0.3$ ${}^{{}^{\circ}}$C	[28]
HeLa	TCSPC system-based FLIM	AP4-FPT, nanogel, polyNNPAM- APTMA- DBThD-AA	No stimulus	$T_{med}=30$ ${}^{{}^{\circ}}$C , $\Delta T_{nuc}=+0.98$ ${}^{{}^{\circ}}$C	[37]
COS-7, HeLa	TCSPC system-based FLIM	FPT, polyNNPAM- SPA- DBD-AA	No stimulus	$T_{cyto}=33.1$ ${}^{{}^{\circ}}$C , $\Delta T_{nuc}=+0.96$ ${}^{{}^{\circ}}$C , $\Delta T_{centro}=+0.7$ ${}^{{}^{\circ}}$C (COS-7) $\Delta T_{nuc}=+0.70$ and −0.03 ${}^{{}^{\circ}}$C in G1 and S/G2 cell cycle phase, respectively (COS-7) $\Delta T_{nuc}=+0.52$ ${}^{{}^{\circ}}$C , $\Delta T_{centro}=+0.82$ ${}^{{}^{\circ}}$C (HeLa) Nuclei and centrosomes bind more FPT than cytosol	[39]
MOLT-4 and HEK293T cells	Flow cytometry, confocal microscopy (MOLT-4) Confocal spectrofluorometry (HEK293T)	1, NNPAM-APTMA-DBThD-AA-BODIPY-AA	No stimulus	Cell cycle-dependent sensor accumulation (MOLT-4) $T_{cyto}=32$ ${}^{{}^{\circ}}$C , $\Delta T_{nuc}=+1$ ${}^{{}^{\circ}}$C (HEK293T)	[40]
BA	TCSPC system-based FLIM	NTG, nucleus thermo green, BODIPY-Hoechst 33258 fusion	1 mM ISO (50 min)	$T_{med}=37$ ${}^{{}^{\circ}}$C , $\Delta T_{nuc}=-0.03\pm 0.87$ ${}^{{}^{\circ}}$C	[35]

Notes: *—MOLT-4, human acute lymphoblastic leukemia. **—gTEMP, fluorescent proteins Sirius and monomeric T-Sapphire linked by self-cleavable Thosea asigna virus 2A peptide. ***—$T_{med}$, $T_{cyto}$, $T_{nuc}$, and $T_{centro}$, temperature of the medium, cytosol, nuclei, and centrosomes, respectively; $\Delta T_{nuc}=T_{nuc}-T_{cyto}$; $\Delta T_{centro}=T_{centro}-T_{cyto}$.

## Data Availability

Not applicable.

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
