# Peer review of "Warm Cells, Hot Mitochondria: Achievements and Problems of Ultralocal Thermometry"

_ijms, 2023, doi:10.3390/ijms242316955_

Round 1

Reviewer 1 Report

Comments and Suggestions for Authors

The proposed review comes at just the right time because questions about the methods used to measure temperatures at micro- or even nanometric scales are now being raised acutely. This is all the more relevant since the value of these methods (and their results) to measure intracellular temperatures is obviously of enormous importance given, as the authors emphasize, the role of temperature on all physiological phenomena is crucial whatever the scales, including the cellular and intracellular scale. The scope covered by the manuscript is considerable and the authors' attempt to take so many aspects (from methods to physiology) into consideration deserves to be highlighted.

As such, although the authors have been generally cautious in this review, there are a few assertions or shortcuts that perhaps should be revisited. For example, when the authors talk about cellular organelles playing a central role in the use of energy (heat in particular), namely the mitochondria, it would be good to emphasize that these are extraordinarily different from one species to another, from one tissue to another, often very heterogeneous in the same cell and that this heterogeneity could exist between the cristae inside a mitochondrion. I'm sure the authors are aware of all this. But on the one hand this should be indicated with much more force in the introduction to the part on the temperature in the cells, the mitochondria and on the other hand this prohibits the comparison of studies which have not been carried out on the same cells or under controlled conditions. 

Particular points

In the manuscript it is written that MTY traces membrane potential. However, we know that the MTY behaves completely differently depending on the cells. Using HeLa or HEK cells, once localized in the mitochondria, it is completely insensitive to the membrane potential. Conversely in skin fibroblasts, MTY tested (under strictly similar conditions) is quite sensitive to membrane potential. This can be illustrated by the opposite effects of oligomycin between the two cell types (see Chrétien et al. 2020 Chemosensors 8:124). 

It might be emphasized in the manuscript that studies are often very partial, questioning their conclusions.

For example, very few studies check the actual respiration of cells simultaneously to measuring their temperature, while we know it is the determining factor controling cell temperature. How to conclude on temperature without knowing if cell respire and at which rate. In the same way, how it is not possible to conclude about the effects of temperature on functions/structures (for example on the state of the super complexes of mitochondria) when the temperature of the mitochondria has not been actually measured?

In conclusion, this is an interesting review which only requires few improvements. 

The English should be carefully checked. In the introduction : « Conductance of biochemical reaction » : what does it mean ? 

« … least harmful … » : less harmful ?

« The resolving the problem of… »

Comments on the Quality of English Language

There are some minor problems, the english should be checked

Author Response

Response to Reviewer 1

 The proposed review comes at just the right time because questions about the methods used to measure temperatures at micro- or even nanometric scales are now being raised acutely. This is all the more relevant since the value of these methods (and their results) to measure intracellular temperatures is obviously of enormous importance given, as the authors emphasize, the role of temperature on all physiological phenomena is crucial whatever the scales, including the cellular and intracellular scale. The scope covered by the manuscript is considerable and the authors' attempt to take so many aspects (from methods to physiology) into consideration deserves to be highlighted.

As such, although the authors have been generally cautious in this review, there are a few assertions or shortcuts that perhaps should be revisited. For example, when the authors talk about cellular organelles playing a central role in the use of energy (heat in particular), namely the mitochondria, it would be good to emphasize that these are extraordinarily different from one species to another, from one tissue to another, often very heterogeneous in the same cell and that this heterogeneity could exist between the cristae inside a mitochondrion. I'm sure the authors are aware of all this. But on the one hand this should be indicated with much more force in the introduction to the part on the temperature in the cells, the mitochondria and on the other hand this prohibits the comparison of studies which have not been carried out on the same cells or under controlled conditions. 

A1. To address this issue, we added the following text in the mitochondrial section:

“Maximum heat production by mitochondria is limited by their maximum respiratory capacity, which, in turn is determined by the amount of respiratory complexes per cristae and the cristae density (Bennett CF, Latorre-Muro P, Puigserver P. Mechanisms of mitochondrial respiratory adaptation. Nat Rev Mol Cell Biol. 2022 Dec;23(12):817-835. doi: 10.1038/s41580-022-00506-6.; Nielsen J, Gejl KD, Hey-Mogensen M, Holmberg HC, Suetta C, Krustrup P, Elemans CPH, Ørtenblad N. Plasticity in mitochondrial cristae density allows metabolic capacity modulation in human skeletal muscle. J Physiol. 2017 May 1;595(9):2839-2847. doi: 10.1113/JP273040.). The maximum respiratory capacity is highly (severalfold) variable in mitochondria of the same tissues of different biological species (Block BA. Thermogenesis in muscle. Annu Rev Physiol. 1994;56:535-77. doi: 10.1146/annurev.ph.56.030194.002535.; Paital B, Samanta L. A comparative study of hepatic mitochondrial oxygen consumption in four vertebrates by using Clark-type electrode. Acta Biol Hung. 2013 Jun;64(2):152-60. doi: 10.1556/ABiol.64.2013.2.2.; Long J, Xia Y, Qiu H, Xie X, Yan Y. Respiratory substrate preferences in mitochondria isolated from different tissues of three fish species. Fish Physiol Biochem. 2022 Dec;48(6):1555-1567. doi: 10.1007/s10695-022-01137-6.), of different tissues of the same species (Long J, Xia Y, Qiu H, Xie X, Yan Y. Respiratory substrate preferences in mitochondria isolated from different tissues of three fish species. Fish Physiol Biochem. 2022 Dec;48(6):1555-1567. doi: 10.1007/s10695-022-01137-6.; ), of the same tissues and species at different metabolic demands (Latorre-Muro P, O'Malley KE, Bennett CF, Perry EA, Balsa E, Tavares CDJ, Jedrychowski M, Gygi SP, Puigserver P. A cold-stress-inducible PERK/OGT axis controls TOM70-assisted mitochondrial protein import and cristae formation. Cell Metab. 2021 Mar 2;33(3):598-614.e7. doi: 10.1016/j.cmet.2021.01.013.). (Moreover, the respiratory capacity of different cristae of the same mitochondrion may differ considerably (Wolf DM, Segawa M, Kondadi AK, Anand R, Bailey ST, Reichert AS, van der Bliek AM, Shackelford DB, Liesa M, Shirihai OS. Individual cristae within the same mitochondrion display different membrane potentials and are functionally independent. EMBO J. 2019 Nov 15;38(22):e101056. doi: 10.15252/embj.2018101056.).) The mitochondria of BA cells, for instance, have a higher content of ETC complexes and cristae density and, thus, higher respiratory and heat-producing capacities than epithelial cells (Nedergaard J, Cannon B. Brown adipose tissue as a heat-producing thermoeffector. Handb Clin Neurol. 2018;156:137-152. doi: 10.1016/B978-0-444-63912-7.00009-6.). Since in the majority of studies the maximum respiratory capacity of cells was not specifically studied, the data in Table 4 are grouped by cell types.” 

Particular points

In the manuscript it is written that MTY traces membrane potential. However, we know that the MTY behaves completely differently depending on the cells. Using HeLa or HEK cells, once localized in the mitochondria, it is completely insensitive to the membrane potential. Conversely in skin fibroblasts, MTY tested (under strictly similar conditions) is quite sensitive to membrane potential. This can be illustrated by the opposite effects of oligomycin between the two cell types (see Chrétien et al. 2020 Chemosensors 8:124). 

It might be emphasized in the manuscript that studies are often very partial, questioning their conclusions.

For example, very few studies check the actual respiration of cells simultaneously to measuring their temperature, while we know it is the determining factor controling cell temperature. How to conclude on temperature without knowing if cell respire and at which rate. In the same way, how it is not possible to conclude about the effects of temperature on functions/structures (for example on the state of the super complexes of mitochondria) when the temperature of the mitochondria has not been actually measured?

A2. The following explanatory remarks were added to the corresponding section:

It should be mentioned that, in several cancer cell lines, in contrast to skin fibroblasts, MTY was found to become ΔΨm-independent after the accumulation by mitochondria (Chrétien, D.; Bénit, P.; Leroy, C.; El-Khoury, R.; Park, S.; Lee, J.Y.; Chang, Y.-T.; Lenaers, G.; Rustin, P.; Rak, M. Pitfalls in Monitoring Mitochondrial Temperature Using Charged Thermosensitive Fluorophores. Chemosensors  2020, 8, 124. https://doi.org/10.3390/chemosensors8040124). The mechanism underlying this phenomenon is not entirely clear. The authors believe that this may be connected with the covalent binding of MTY to the matrix proteins. Another possible reason is the effect of pH on the ΔΨm-dependence of MTY distribution. Indeed, the amide group of MTY should be protonated by 85% (calculation using the ACD/Labs software) at normal cytosolic pH (~7.2) and predominantly deprotonated at the normal pH of the mitochondrial matrix (7.8-8.0), thus losing the ΔΨm-dependence. Hence, H+ buffering with the Pi/adenine nucleotide system and the modulation of this buffering by Pi transport and ATP synthesis/hydrolysis should affect the matrix pH and ΔΨm-dependence of MTY. Therefore, the application of sensors with a high ΔΨm- or pH-sensitivity makes the calibration of the mitochondrial thermal response quite risky. Nevertheless…

In conclusion, this is an interesting review which only requires few improvements. 

The English should be carefully checked. In the introduction : « Conductance of biochemical reaction » : what does it mean ? 

« … least harmful … » : less harmful ?

« The resolving the problem of… »

A3. In the revised version of the manuscript, English was edited by a professional translator.

Reviewer 2 Report

Comments and Suggestions for Authors

Kruglov et al. provide a detailed review of the role of temperature in cells for regulating various physiological processes.  They focus on mitochondria as the main source of cellular thermogenesis.  Furthermore, they discuss various tools and techniques for quantifying temperature at the cellular and organellar levels.  Importantly, they also consider the strengths and limitations of different approaches.

With the focus on mitochondria, it would be informative to discuss what is known about the effects of temperature on membrane architecture.  For example, reports indicate that treatment with uncouplers, such as FCCP, result in a decrease in cristae density.  What is known about the effects on mitochondrial membranes following relatively large changes in temperature?  In this respect, what may be the role of the various heat shock (or cold shock) proteins in maintaining mitochondrial integrity during temperature challenges?

Overall, this review is thoughtful and thorough and will likely be an asset to scientists interested in the role of temperature in the cell.

Minor corrections:

Line 39:  “The resolving the” is a typo.

Line 192:  “phonon” should be “photons”

Line 248: “ways on how to measure” should be “ways to measure”

There are numerous other grammatical errors throughout, which the authors should correct. 

Comments on the Quality of English Language

There are miscellaneous grammatical errors that should be addressed.  But, overall, the English is comprehensible.

Author Response

Response to Reviewer 2

Kruglov et al. provide a detailed review of the role of temperature in cells for regulating various physiological processes.  They focus on mitochondria as the main source of cellular thermogenesis. Furthermore, they discuss various tools and techniques for quantifying temperature at the cellular and organellar levels.  Importantly, they also consider the strengths and limitations of different approaches.

With the focus on mitochondria, it would be informative to discuss what is known about the effects of temperature on membrane architecture.  For example, reports indicate that treatment with uncouplers, such as FCCP, result in a decrease in cristae density.  What is known about the effects on mitochondrial membranes following relatively large changes in temperature?  In this respect, what may be the role of the various heat shock (or cold shock) proteins in maintaining mitochondrial integrity during temperature challenges?

A1. To highlight the mentioned by the Reviewer aspects related to the effect of temperature on mitochondrial processes, we have included an additional section in the review: 3.6. Avoidance of thermal damage.

Overall, this review is thoughtful and thorough and will likely be an asset to scientists interested in the role of temperature in the cell.

Minor corrections:

Line 39:  “The resolving the” is a typo.

Line 192:  “phonon” should be “photons”

Line 248: “ways on how to measure” should be “ways to measure”

There are numerous other grammatical errors throughout, which the authors should correct.

A2. In the revised version of the manuscript, English was edited by a professional translator.